

# Modeling Trans-Pacific Transport using Hemispheric CMAQ during April 2010: Part 1. Model Evaluation and Air Mass Characterization for the Estimation of Stratospheric Intrusion on Tropospheric Ozone

Syuichi Itahashi[1], Rohit Mathur[2], Christian Hogrefe[2], and Yang Zhang[3]

[1] Environmental Science Research Laboratory, Central Research Institute of Electric Power Industry (CRIEPI), 1646 Abiko, Abiko, Chiba 270–1194, Japan
[2] Environmental Protection Agency (EPA), Computational Exposure Division, National Exposure Research Laboratory, Office of Research and Development, Research Triangle Park, NC 27711, U.S.A.
[3] Department of Marine, Earth, and Atmospheric Sciences (MEAS), North Carolina State University (NCSU), Campus Box 8208, Raleigh, NC 27695, U.S.A.

*Correspondence to*: Syuichi Itahashi (isyuichi@criepi.denken.or.jp)

**Abstract.**

Trans-Pacific transport has been recognized as a potential source of air pollutants over the U.S.A. The state-of-the-science
Community Multiscale Air Quality (CMAQ) Modeling System has recently been extended for hemispheric-scale modeling applications (referred to as H-CMAQ). In this study, H-CMAQ is applied to study the trans-Pacific transport during April 2010. The results will be presented in two continuous papers. In this part 1 paper, model evaluation for tropospheric ozone ($O_3$) is presented. Observations at the surface, by ozonesondes and airplane, and by satellite across the northern hemisphere are used to evaluate the model performance for $O_3$. H-CMAQ is able to capture surface and boundary layer (defined as surface
to 750 hPa) $O_3$ with a normalized mean bias (NMB) of -10%; however, a systematic underestimation with an NMB up to -30% is found in the free troposphere (defined as 750-250 hPa). The surface and aloft relative humidity (RH) showed a positive bias around NMB of +10% or greater. In addition, a new air mass characterization method is developed to distinguish influences of stratosphere-troposphere transport (STT) from the effects of photochemistry on $O_3$ levels. Potential vorticity (PV) is used to diagnose air masses of stratospheric origin and related to RH in order to characterize stratospheric air masses.
The tropopause location is determined using a PV threshold value of 2.0 PVU (1 PVU = $10^{-6}$ m$^2$ K kg$^{-1}$ s$^{-1}$). The constructed PV-RH relationship indicates that PV of 2.0 PVU generally corresponds to RHs of 30-40%. The air mass characterization method is then developed based on the ratio of $O_3$ and an inert tracer indicating stratospheric $O_3$ to examine the importance of photochemistry, and the PV-RH relationship is used to determine stratospheric intrusions. Over the U.S.A., STT impacts show large day-to-day variations, and STT impacts can either originate from the same air mass over the entire U.S.A. with an
eastward movement, or stem from different air masses at different locations. The relationship between surface $O_3$ mixing ratios and estimated stratospheric air masses in the troposphere show a negative slope, indicating that high surface $O_3$ values are primarily affected by other factors (i.e., emissions), whereas this relationship shows an almost flat slope at elevated sites,



indicating that STT has a near constant impact at elevated sites. Based on this newly established air mass characterization technique, this study can contribute to understand the role of STT, and also the implied importance of emissions leading to high surface $O_3$. Further research focused on emissions is discussed in a subsequent part 2 paper.

# 1 Introduction

Tropospheric ozone ($O_3$) is a secondary air pollutant produced by a chain of reactions involving photochemical oxidation of volatile organic compounds (VOCs) in the presence of nitrogen oxides ($NO_x$) (Haagen-Smit and Fox, 1954). Ozone plays a key role in tropospheric chemistry by controlling the oxidizing capacity through the production of hydroxyl (OH) radicals, and is an important greenhouse gas throughout the troposphere (Logan, 1985). Ground level $O_3$ poses significant risks to human health and therefore many countries regulate it as a criteria pollutant with an ambient air quality standard. In
the U.S.A., the National Ambient Air Quality Standard (NAAQS) for $O_3$ is based on the annual 4[th] highest maximum daily 8-h concentration (MD8O3) averaged over three years and its threshold values have been decreasing from 80 ppbv in 1997 to 75 ppbv in 2008, and 70 ppbv in 2015 (EPA, 2018). Long-term trends of rural $O_3$ during 1990-2010 revealed significant $O_3$ decreases in the eastern U.S.A. during spring and summer whereas no significant $O_3$ decrease was found in the western U.S.A. during spring (Cooper et al., 2012). Analysis of trends in surface $O_3$ levels between 1998 and 2013 showed that the highest $O_3$
concentration in the U.S.A. has been reduced in response to substantial decline of precursor (Simon et al., 2015). It was also shown that the low $O_3$ days have increased and led to the narrowing of the $O_3$ concentration range across the U.S.A.

From the viewpoint of global air quality changes, the recent acceleration of anthropogenic emissions in East Asia (Itahashi et al., 2013, 2014, 2015), may impact atmospheric composition at not only the local and regional scales, but also the global scale. By combining trajectory analysis with detailed chemical and meteorological data, it was suggested that the
20 emissions were lifted into the free troposphere over Asia and then transported to North America in about 5-8 days (Jaffe et al., 1999). Trans-Pacific transport has been studied over the past decade because of its potential impact on rising background $O_3$ concentrations (Cooper et al., 2010). Asian contributions to surface $O_3$ levels in the U.S.A. require an additional challenge to meeting more stringent NAAQS for $O_3$ (Fiore et al., 2002). A typical case of trans-Pacific transport occurred during the so-called "perfect dust storm" during April 2001, transporting Asian dust to North America (Huebert et al., 2003). From an air
pollutant perspective, it was reported that the impact of Asian emissions increased background concentration of $O_3$ by 1 ppbv (2.5%) on a monthly average basis and up to 2.5 ppbv on a daily average basis over the western U.S.A. in April 2001 (Wang et al., 2009). Background $O_3$ levels entering western North America in spring have increased by approximately 10 ppbv between 1984 and 2002 based on a compilation of observations over the west coast of the U.S.A., and the possible cause for this increase was thought to be Asian emission trends (Jaffe et al., 2003). Asian air pollution can enhance surface $O_3$ mixing
ratios by 5-7 ppbv over western North America in April-May 2006, and the doubled Asian anthropogenic emissions increase during 2000-2006 was estimated to have the impact by 1-2 ppbv (Zhang et al., 2008). The tripling of Asian anthropogenic emissions from 1985 to 2010 increased $O_3$ mixing ratios by 2-6 ppbv in the western U.S.A. and by 1-3 ppb in the eastern



U.S.A. on a monthly-mean basis, with the maximum effect occurring in April-June; this increase was suggested to more than offset the benefits of 25% domestic reduction in the western U.S.A. (Jacob, et al., 1999).

The occurrence of trans-Pacific transport can be inferred from variations in the jet stream related to La Niña and El Niño. The springtime Asian outflow may be enhanced following an El Niño winter due to the eastward extension of the atmospheric circulation over the Pacific-North America sector and the southward shift of the subtropical jet stream (Koumoutsaris et al., 2008; Lin et al., 2015). According to the NOAA Climate Prediction Center (CPC), 2009–2010 wintertime was influenced by strong El Niño conditions (NOAA, 2018). Because of the favorable condition for trans-Pacific transport, it was reported that Asian dust reached North America on at least five occasions during April 2010 (Uno et al., 2011). During May-June 2010, the Asian enhancement to MD8O3 in the western U.S.A. was estimated to reach 8-15 ppbv in high-elevation regions during strong trans-Pacific transport events (Lin et al., 2012a).

Another process affecting tropospheric $O_3$ is stratosphere-to-troposphere transport (STT), which is known to be a significant contributor to the tropospheric $O_3$ budget (Lelieveld and Dentener, 2000). The tightening of the $O_3$ NAAQS and a continuous decrease of anthropogenic emissions have led to an increased focus on STT. On one hand, stratospheric intrusion of $O_3$ was found to be below 20 ppbv during March-October 2001 over the entire U.S.A (Fiore et al., 2003), on the other hand, a total of thirteen events were identified during April-June 2010 when stratospheric intrusion impacts reached 20-40 ppbv while accounting for 50-60% of total $O_3$ at fifteen high-elevation (> 1.4 km above mean sea level; m.s.l.) sites in the western U.S.A. (Lin et al., 2012b). From the perspective of interannual variability, springtime stratospheric intrusions may be enhanced following a La Niña winter due to a meandering of the jet stream, and a large variability in terms of magnitude and frequency have been shown from 1990 to 2012 (Lin et al., 2015). The fraction of $O_3$ in the troposphere that originates from the stratosphere is still uncertain due to its strong dependence on season and location which affect tropopause heights and is therefore still an area of active research (Mathur et al., 2017).

Table 1 summarizes these studies that provided the motivation for evaluating the impacts of both precursor emissions and STT on tropospheric $O_3$. April 2010 is selected as the study period because enhancement of trans-Pacific transport is expected during the 2009-2010 El Niño winter. Along with the gradual reduction of precursor emissions of $NO_x$ and VOCs in the U.S.A., a gradual decrease of MD8O3 mixing ratios can be expected and showed a decreasing trend by 0.4%/year; however, mean MD8O3 mixing ratios in 2010 showed a local maximum and the number of NAAQS threshold exceedances was larger than usual as shown in Fig. S1 in the supplemental material. This period has already been the subject of other studies (e.g., Uno et al., 2011; Lin et al, 2012a), however, the methods used in this study to investigate the impacts of trans-Pacific transport differ from previous studies. The objective of this research is to better understand the relative contributions of precursor emissions from East Asia and the U.S.A. because the trans-Pacific transport has been recognized as an important factor. Previous studies primarily focused on Asian impacts on the western U.S.A., while this study investigates impacts across the entire U.S.A. In addition, some stratospheric intrusion events have been reported during spring 2010 (Lin et al., 2012b), therefore this period is suitable to examine not only trans-Pacific transport but also stratospheric intrusion. This study will contribute to improving our understanding of the importance of both emission impacts and stratospheric intrusion of $O_3$. The



results will be presented in two parts. In this part 1, we present the model evaluation and introduce a new method to identify and characterize periods during which surface and lower tropospheric $O_3$ may be influenced by stratospheric intrusions. This manuscript is organized as follows. In section 2, the modeling system and simulation set up are described, details on the surface, ozonesonde, airplane, and satellite observations used to evaluate the model performance are presented, and evaluation

protocols are defined. In section 3, the analysis of model results and comparisons with observations are documented and the newly developed air mass characterization method is introduced and applied to investigate stratospheric intrusions. Finally, the conclusion section includes limitations of this work, future perspectives, and a brief introduction to part 2. In Part 2, the contributions of emissions leading to higher $O_3$ mixing ratio will be presented in a future paper.

## 2 Methodology

### 2.1 Modeling System and Simulation Set Up

The model used in this work is the Community Multiscale Air Quality (CMAQ) version 5.2 extended for hemispheric applications (H-CMAQ) (Mathur et al., 2017). To investigate the impact of emissions from East Asia, H-CMAQ is configured to cover the entire Northern Hemisphere, utilizing a horizontal discretization of 187×187 grid points with a grid spacing of

108 km. The terrain-following vertical coordinate utilizes 44 layers of variable thickness to resolve the model vertical extent between the surface and 50 hPa based on the extension of the previous 35 layers system (Mathur et al., 2017). The revised layer structure using 44 layers with significantly finer resolution above the boundary layer (BL) better represents long-range transport in the free-troposphere (FT) as well as STT processes, and influences from cloud mixing on both the sub-grid and resolved scales. The emission inputs are based on the Hemispheric Transport of Air Pollution version 2 (HTAP2) modeling

experiments, and the detailed description can be found in previous studies (Janssens-Maenhout et al., 2015; Pouliot et al., 2015; Galmarini et al., 2017; Hogrefe et al., 2018). For gas-phase chemistry, cb05e51 is used (Appel et al., 2017). This gas-phase mechanism includes the condensed halogen chemistry that leads to $O_3$ loss in marine environments (Sarwar et al., 2015). For aerosol chemistry, aero6 with nonvolatile primary organic aerosol (POA) (Simon and Bhave, 2012) is adopted.

Potential vorticity (PV) has been shown to be a robust indicator of air mass exchange between the stratosphere and

the troposphere. The value of PV generally increases with altitude, and previous studies suggested that a value of 2 PVU (1 $PVU = 10^{-6} \ m^2 \ K \ kg^{-1} \ s^{-1}$) is an indicator of stratospheric air (Hoskins et al., 1985; Wernli and Bourqui, 2002; Itoh and Narazaki, 2016). PV shows a strong positive correlation with $O_3$ (Danielsen, 1968), and modeling studies have used this correlation to develop scaling factors that specify $O_3$ in the modeled upper troposphere/lower stratosphere (UTLS) based on estimated PV. The reported $O_3$/PV ratios exhibited a wide range from 20 to 100 ppbv/PVU with differences in locations, altitude and season

(e.g., Ebel et al, 1991; Carmichael et al, 1998; McCaffery et al, 2004). To account for the seasonal, latitudinal and altitude dependencies in the $O_3$-PV relationship, a dynamic $O_3$-PV function was developed (Xing et al., 2016) to consider latitude, altitude, and time based on 21-year ozonesonde records from the World Ozone and Ultraviolet Radiation Data Centre



(WOUDC) and corresponding PV values from WRF-CMAQ simulations across the northern hemisphere from 1990 to 2010 and is used in H-CMAQ (Mathur et al., 2017). To track stratospheric air masses, the $O_3$ estimated using the O3/PV relationship is added as a chemically-inert tracer species in the H-CMAQ simulations. The O3PV tracer undergoes the same transport, scavenging, and deposition processes as $O_3$, but its mixing ratios are not affected by chemical production or loss processes.

The meteorological fields are simulated by the Weather Research and Forecasting (WRF) model version 3.6.1 using the same vertical configuration as H-CMAQ. WRF simulation started from 1 March 2009 with more than one year of spin-up time prior to the analysis period of April 2010. The WRF model is configured to use the rapid radiative transfer model for global climate models (RRTMG) radiation scheme for both longwave and shortwave (Iacono et al., 2008), Morrison double-moment scheme (Morrison et al., 2009) and Grell convective parameterization (Grell 1993; Grell and Devenyi, 2002) for

microphysics and cumulus parameterization, and Mellor-Yamada-Janjic scheme for planetary boundary layer (Janjic et al., 1994). Wind, temperature, and water vapor fields are nudged towards NCEP/NCAR final analysis data for all layers, these analysis data have 1 degree spatial and 6 h temporal resolution (NCEP, 2018). The WRF meteorological fields are converted to the format required by H-CMAQ using MCIP version 4.3 (Otte and Pleim, 2010), and then used for the H-CMAQ simulation. Relative humidity (RH) can also be used to diagnose stratospheric air masses because the stratosphere is characterized by dry

air. CMAQ used the meteorological fields simulated by WRF and calculated RH based on the improved Magnus form approximation for saturation vapor pressure (Alduchov and Eskridge, 1996), and internally set the maximum value on 99% and minimum value on 0.5%. The CMAQ simulation started from 1 March 2010 and initialized with three-dimensional chemical fields from prior model simulations for 2010 by Hogrefe et al. (2018); March is discarded as a spin-up period and April is used as the analysis period.

## 2.2 Observations and Evaluation Protocols

### 2.2.1 Ground-based Surface $O_3$ Observations

        The northern hemispheric modeling domain and ground-based observations used in this study are shown on the map in Fig. 1. Global ground-based surface $O_3$ observations were obtained from the World Data Centre for Greenhouse Gases

(WDCGG; shown as red circles in Fig. 1). For the study period of April 2010, this dataset contained 52 sites in North America, Europe, and several remote locations with only limited coverage over Asia (WDCGG, 2018). To overcome this limitation, surface $O_3$ observations are also obtained from the Acid Deposition Monitoring Network in East Asia (EANET) program which provides measurements at 12 sites in Japan, 3 sites in the South Korea, 1 site in Russia, and 4 sites in Thailand. However, the observed data are only available on a daily-mean basis for Russia, and a monthly-mean basis for South Korea and Thailand.

Therefore, the only EANET monitors used in this study (EANET, 2018) are those located in Japan; these 9 sites with available data for April 2010 are shown as green triangles in Fig 1. In addition, surface $O_3$ observations over the U.S.A. were obtained from the Clean Air Status and Trends Network (CASTNET) and are shown as blue squares in Fig. 1. CASTNET monitors (CASTNET, 2018) are located mostly in rural and remote areas, which makes them appropriate for comparison to $O_3$ fields





from the coarse resolution H-CMAQ simulations. CASTNET data are available at 81 sites during April 2010. MD8O3 values for April 2010 are calculated from the hourly observations at these WDCGG, EANET, and CASTNET stations.

### 2.2.2 Ozonesondes

An evaluation of simulated vertical $O_3$ profiles is needed to analyze the model's ability to capture the behavior of aloft $O_3$. To this end, we obtained ozonesonde data distributed by the WOUDC as well as additional ozonesonde soundings available over the U.S.A. and Greenland that are collected and distributed by the National Oceanic and Atmospheric Administration Earth System Research Laboratory (NOAA-ESRL) (NOAA, 2018a). The total number of available ozonesonde sites during April 2010 was 33 (locations shown as yellow stars in Fig. 1). The data for Hilo and Boulder are available in both the WOUDC and NOAA-ESRL database; the NOAA-ESRL data are used because they include information on uncertainties of the $O_3$ measurements. Detailed information for each site, including country, site name, latitude (°N), longitude (°E), elevation (m a.s.l.), and the number of launches during April 2010, is provided in Table 2. There are 6 sites located in the U.S.A., 10 sites in Canada, 5 sites over Asia, and 12 sites over the Europe. In addition to measured $O_3$ mixing ratios, observed RH vertical profiles are used to evaluate the model performance.

### 2.2.3 Airplane

In addition to ozonesonde data to evaluate the vertical $O_3$ distribution, observations from research aircraft for three sites located in the U.S.A. (Cape May, New Jersey; Homer, Illinois; and Southern Great Plains, Oklahoma) are available from NOAA-ESRL (NOAA, 2018b) for April 2010. Because the observations at Cape May and Homer are only available for a single day during April 2010, we only used the NOAA-ESRL aircraft data at Southern Great Plains which is shown as a gray diamond in Fig. 1. A total of seven flights were conducted at this site during April 2010. In addition to $O_3$ mixing ratios, RH was used to evaluate the model performance.

### 2.2.4 Satellite

Tropospheric column $O_3$ observed by the Ozone Monitoring Instrument (OMI) onboard the National Aeronautics and Space Administration (NASA) Earth Observing System Aura satellite is used in this study. The methodology to estimate the tropospheric column has been developed (Ziemke et al., 2006) and consists of taking the differences between total column $O_3$ observed by OMI and stratospheric column $O_3$ observed by the Microwave Limb Sounder (MLS). The monthly-mean tropospheric $O_3$ column data are available between 60°S and 60°N (NASA, 2018a). Because this tropospheric column $O_3$ data are monthly-mean data, in order to take into account the daily missing data by OMI, total column data (OMTO3d) are utilized to obtain the information on daily missing data in order to compare with the model (NASA, 2018b). This total column data are the products of averaging only good quality flag of level-2 swath data and then gridded into 1×1 degree. Such an approach



considering daily deficit data has also been applied in previous study (e.g., Chatani et al., 2014). To diagnose the tropopause in the model, PV with a value of 2 PVU is used as threshold. This diagnosis is applied above the boundary layer to avoid the misdiagnosis near the surface due to the high value of PV caused by turbulence.

5    **2.2.5 Evaluation Protocol**

To evaluate model performance, the Pearson's correlation coefficient (R) with student's *t*-test is used for assessing the statistical significance level. The normalized mean bias (NMB) and the normalized mean error (NME) are calculated using the following equations (e.g., Zhang et al., 2006);

$$R = \frac{\sum_1^N |(O_i - \bar{O})(M_i - \bar{M})|}{\sqrt{\sum_1^N (O_i - \bar{O})^2} \sqrt{\sum_1^N (M_i - \bar{M})^2}} \qquad (1)$$

$$NMB = \frac{\sum_1^N (M_i - O_i)}{\sum_1^N O_i} \qquad (2)$$

$$NME = \frac{\sum_1^N |M_i - O_i|}{\sum_1^N O_i} \qquad (3)$$

where, N is the total observation number, $O_i$ and $M_i$ represent each individual observation and model value respectively, and $\bar{O}$ and $\bar{M}$ represent the arithmetical mean of observations and model values respectively. Based on a compilation of model evaluation reports, Emery et al. (2017) suggested threshold values of R > 0.75, NMB < ±5%, and NME < 15% as performance

15    goal, and threshold values of R > 0.50, ±5% < NMB < ±15%, and 15% < NME < 25% as performance criteria for 1-hr O₃ or MD8O3 simulated by regional-scale air quality models. Although these recommendations were developed for regional-scale air quality models and suggested to apply over time-space averaging scales of no longer than one month and no more than 1000 km, these three criteria are applied in this work to judge the performance of the April 2010 H-CMAQ simulations due to the lack of other commonly-accepted model performance criteria for hemispheric or global scale O₃ simulations. Evaluation

20    of surface O₃ simulated by global models indicated a somewhat loose threshold might be required because of the use of a coarse grid resolution (Zhang et al., 2012; He et al., 2015a, 2015b).



## 3 Simulation Results and Discussion

### 3.1 Model Evaluation

A scatterplot of modeled vs. observed MD8O3 at WDCGG, EANET, and CASTNET, sites during April 2010 is shown in Fig. 2 using colors and symbols that are consistent with Fig. 1. A summary of the statistical analysis is provided in Table 3. Almost all of the EANET (green triangles) and CASTNET (blue squares) MD8O3 data pairs were within the 1:2 lines across the entire $O_3$ mixing ratio range. The comparison of H-CMAQ values with EANET observations over Asia shows an R value of 0.49 which is statistically significant at a level of $p < 0.001$, an NMB of −12.6% and an NME of 20.6% (Table 3). A comparison to CASTNET observations over the U.S.A. shows that the mean observed and modeled values are close with an NMB of -0.9%, and an NME of 12.6%, and that R had a value of 0.61 with $p < 0.001$. This indicates that the H-CMAQ simulations captured the CASTNET observational data within the model criteria performance suggested by Emery et al. (2017). A comparison to WDCGG data across the Northern Hemisphere shows an R of 0.49 with $p < 0.001$, an NMB of −19.3%, and an NME of 23.7%. The mean model value is approximately 10 ppbv less than the mean of the observations. This feature is also evident in the scatter-plot shown in Fig. 2. While observed values reach more than 100 ppbv of MD8O3, the corresponding model values are only about half of these high MD8O3 mixing ratios, indicated by a clustering of WDCGG pairs on the 2:1 line. To investigate this model underestimation further, the spatial distributions of monthly mean modeled and observed $O_3$ mixing ratios are examined in Fig. 3 which shows observed high mixing ratios over eastern Europe. In particular, the four WDCGG monitors at Kosetice, Czech Republic (15.08°E, 49.58°N, 534 m a.s.l.), K-puszta, Hungary (19.55°E, 46.97°N, 125 m a.s.l.), Rucava, Latvia (21.17°E, 56.16°N, 18 m a.s.l.), and Zoseni, Latvia (25.54°E, 57.08°N, 182 m a.s.l.) measured MD8O3 mixing ratios larger than 100 ppbv by 14, 10, 5, and 4 days, respectively, during April 2010. An analysis of data collected during an airborne measurement campaign during 15-18, April 2010 over Siberia reported enhanced $O_3$ mixing ratios influenced by long-range transport, biomass burning plumes, and stratospheric intrusion (Berchet et al., 2013). Since the biomass burning emissions used in the current H-CMAQ simulations are based on climatological averages rather than year-specific events, the model underestimation may at least partially be due to the representation of these emissions. From the viewpoint of meteorology, the blocking events over European Russia during spring-summer 2010 were reported, and positive anomaly of $O_3$ total column over the regions adjacent to the anticyclones (i.e., Europe) were analyzed (Sitnov et al., 2017). Removing data from these four sites from the analysis yields model performance metrics of an R of 0.63 with $p < 0.001$, an NMB of −14.4%, and an NME of 19.5%; which are comparable to performance at the EANET sites. Aside from the underestimation of high observed MD8O3 mixing ratios at these four European sites, H-CMAQ generally captured the WDCGG observations. Summarizing the model evaluation with surface observations, it is confirmed that model reasonably captures MD8O3 almost within model performance criteria of Emery et al. (2017).

To investigate the vertical profiles of $O_3$, ozonesonde and airplane data are used in this study. In Fig. 4, time-height cross-sections ("curtain plots") of hourly modeled $O_3$, O3PV, and RH values during April 2010 are shown at the location of the ozonesonde sites in the U.S.A., i.e., at Hilo, HI, Trinidad Head, CA, Boulder, CO, Huntsville, AL, Wallops Island, VA,





and Rhode Island, RI. The plots also show contour lines of modeled PV for PV values of 1.0, 1.5, 2.0, 2.5, and 3.0 PVU with the thick lines indicating a value of 2.0 PVU that can be used to diagnose the tropopause. Generally, $O_3$ and O3PV mixing ratios are very similar in the upper layers, especially above the 2.0 PVU line, indicating that $O_3$ mixing ratio in these layers are dominated by stratospheric air mass. Below the tropopause as diagnosed by the PV = 2.0 PVU line, $O_3$ mixing ratios are

generally higher than O3PV mixing ratios, suggesting that $O_3$ was photochemically produced in the troposphere. On the other hand, instances of $O_3$ mixing ratios lower than O3PV mixing ratios are indicative of photochemical loss. One typical example of such photochemical loss can be seen at Hilo (Fig. 4 (a)). At that location, $O_3$ mixing ratios are less than 30 ppbv below 2 km whereas the O3PV mixing ratios are larger than 40 ppbv. A likely driver of this strong $O_3$ loss is the halogen chemistry in marine environments implemented in H-CMAQ (Sarwar et al., 2015) because Hilo site is surrounded by ocean. The impact of

photochemical processes is further discussed in Section 3.2. Ozone mixing ratios at the level of the tropopause as diagnosed by the 2 PVU lines generally are around 100 ppbv (light blue colors in Fig. 4). Although high values of PV are typically seen in the upper layer above 10 km, it should be noted that high PV can occasionally also be found in the lower troposphere where could be associated with convection. RH values are below 10% (white colors in Fig. 4) above the tropopause and steeply increased near or below the tropopause as diagnosed by the 2.0 PVU lines. Based on the rough estimation shown in the curtain

plots of Fig. 4, RH values at the level of the tropopause are typically on the order of 30-40%. The relationship between PV and RH and its application to characterize stratospheric air mass is further discussed in Section 3.2. In Fig. 4, the launch times of available ozonesonde measurements are indicated by yellow stars and we discuss below the comparison of model profiles to measurements from these launches.

        The vertical profiles of observed and modeled $O_3$ and RH, as well as modeled O3PV and PV are shown in Fig. 5. In

this figure, vertical red lines corresponding to a PV value of 2.0 PVU are inserted as index of stratospheric air masses, and the diagnosed stratosphere is colored with purple. A quantitative comparison between simulations and observations is conducted by averaging the observations onto the vertical grid spacing used by the model. The vertical layers are then assigned to three vertical ranges based on typical pressure values, i.e., the boundary layer (surface to approximately 750 hPa), the free troposphere (approximately 750-250 hPa), and the upper model layers (approximately 250-50 hPa) following the same

approach used in our previous study (Hogrefe et al., 2018). Furthermore, the statistical analysis is performed separately for the three regions of U.S.A. and Canada, Asia, and Europe and the three layer ranges defined above. Results of this statistical analysis for $O_3$ mixing ratios are shown in Table 3 and reveal that over all three regions the model performed the best for the boundary layer in terms of NMB and NME. The observed mean boundary layer values of around 45 ppbv over the three regions are well captured by the model. Over the U.S.A. and Canada, model performance in the boundary layer satisfies the

performance criteria for all three metrics of R, NMB, and NME, and over Asia and Europe, NMB and NME also satisfy the performance criteria whereas R is less than 0.5. Compared to the results for the boundary layer, the model tends to underestimate the observed $O_3$ mixing ratios in the free troposphere and the upper model layers. In the free troposphere, the mean observed value is around 80 ppbv while the mean model value is below 60 ppbv. As a result, NMB values are greater than -15% and NME values are greater than 20%. This underestimation is also present in the upper model layers; the mean



observed values of 500 – 1000 ppbv are consistently underestimated by about 100 ppbv by the model across the three regions as shown in Table 3. R values tend to increase from the boundary layer to the free troposphere and the upper model layers due to model's ability to capture the increase of $O_3$ mixing ratios with height. The higher R values in the free troposphere, a region where impacts of photochemistry on $O_3$ variability is smaller, also suggest greater confidence in the model dynamics, which drive $O_3$ variations in this part of the atmosphere. Table S1 in the supplemental material shows the statistical results that are obtained when grouping stations into latitude ranges. The results indicate that model performance is similar to that shown in Table 3 and discussed above. These results suggest that although the revision of the dynamic-PV approach described in Xing et al., 2016 led to improved results compared to earlier implementations of the scaling approach, there is a need for further refinement of the approach to better capture high mixing ratios of stratospheric $O_3$. Using a finer verical resolution for the upper layers and extending the model top beyond 50 hPa could be potential strategies to address this need.

As expected, Fig. 5 shows that RH has higher values and large variations in the troposphere and lower values in the stratosphere. For the analysis of modeled vertical profiles, model results of maximum and minimum values within ±2 hours from observation time are also shown, and the range of RH showed large variations at lower altitude. Table 4 summarizes the statistical analysis divided into the three regions and three vertical domains for RH. It was found that the model generally overestimates RH over all regions and all three layers ranges. Although the NME value seems high for the upper layers, this is caused by the low absolute values of RH. The mean absolute differences between observed and modeled RH values are 1-2 % over the U.S.A. and Canada and Europe, and 8% over Asia. In Table S2 of the supplemental material, the RH results in Table 4 are presented for different latitude bands in the same fashion as the $O_3$ results in Table S1. Results show that model performance is similar to that discussed for Table 3. The systematic positive bias of RH occurs despite using analysis nudging for wind, temperature, and water vapor in the WRF simulations. Positive bias in predicted RH is also found in meteorological simulations performed for AQMEII (Vautard et al., 2012).

The tropopause diagnosed by PV = 2.0 PVU is located around 10-12 km at five ozonesonde sites in the U.S.A. except Hilo where it is located around 16 km. Observations in late April show instances of tropopause heights at or below 6 km (e.g., 27 April at Trinidad Head (Fig. 5 (b)), 29 April at Boulder (Fig. 5 (c)), and 27 April at Huntsville (Fig. 5 (d))). These cases illustrate large impacts of episodic STT, with observed $O_3$ mixing ratios steeply increasing from 100 ppbv at around 6 km to over 500 ppbv at around 8 km The profiles obtained from the H-CMAQ simulations do not capture this steep increase and only show a gradual increase. This finding further supports a need for further refinement of representing stratospheric high $O_3$ mixing ratios as discussed above in the context of Table 3. In terms of RH, observed RH values show a sudden decline from around 60% to near 0% at Trinidad Head and Huntsville, whereas modeled RH values show a gradual decrease with large temporal variations. This contributes to the modeled positive RH bias shown in Table 4.

The comparison of model 3D $O_3$ structure at Southern Great Plains, Oklahoma, with research aircraft measurements is illustrated in Fig. 6. At this site, the curtain plot of modeled $O_3$ is shown for the entire month of April 2010 in the top row and zoomed inserts for the seven observational times indicated by gray diamonds above those plots are shown in the second row with each box showing airplane observations overlaid on H-CMAQ values. For $O_3$, observed and modeled mixing ratios



increased from about 30 ppbv at 1 km to about 55 ppbv at 5 km, except for flight #1 which shows persistently high mixing ratios of 50-60 ppbv throughout this altitude range. However, observed high mixing ratios of $O_3$ over 70 ppbv during flight #5, 6, and 7 are not captured by H-CMAQ. In contrast to $O_3$, observed and modeled RH generally decreased from 1 km to 5 km as shown in rows 3 and 4. Overestimation in model RH is noted for flights #3, 4, and 6 above 3 km. Considering the profiles

of $O_3$ and RH, flight # 6 might be a case of STT because observed RH is less than 10% and observed $O_3$ mixing ratios exceed 75 ppbv; however, the model fails to reproduce this behavior. The profile data averaged over all airplane ascents and descents are plotted in the bottom panel of Fig. 6, and statistical analysis of these profiles is included in Table 3 for $O_3$ and Table 4 for RH. Similar to the evaluation results for ozonesondes, the model could reasonably capture observed $O_3$ and RH profiles, but $O_3$ mixing ratios are generally underestimated and RH is overestimated.

The observed and modeled tropospheric column $O_3$ are compared in Fig. 7. The observed latitudinal gradients in tropospheric column $O_3$ with values greater than 40 D.U. over mid-latitudes, column values around 30 D.U. over high- and low-latitudes, and values below 20 D.U. over the Pacific Ocean near the equator are captured well by H-CMAQ. To illustrate the differences between observations and simulations, the normalized bias is also shown in Fig. 7. This normalized bias map shows model tropospheric column $O_3$ overestimation over Russia and Africa and a slight underestimation over the Pacific

Ocean. While the comparison with surface observations from WDCGG shows model underestimation at four sites over eastern Europe, the, model slightly overestimates tropospheric column $O_3$ in this region. The results of the statistical analysis for tropospheric column $O_3$ are also listed in Table 3. The mean of observed and modeled tropospheric column $O_3$ across Northern Hemisphere is close, with an R of 0.65, an NMB of 4.7%, and an NME of 13.5%. The performance of tropospheric column $O_3$ judged based on the evaluation protocol developed for mixing ratios, suggests that the model satisfies the performance

criteria proposed by Emery et al. (2017).

**3.2 Air Mass Characterization Method**

        In order to characterize whether $O_3$ in a given air mass is dominated by photochemistry or stratospheric intrusion, and further estimate the impacts of STT, a new air mass characterization method is established here. The relationship between PV

and RH is first estimated to characterize stratospheric air mass at ozonesonde observational sites. The scatter-plot between modeled PV and both modeled and observed RH are plotted in Fig. 8 (a) and (b). The colors in the scatter-plots indicate modeled or observed $O_3$ mixing ratios. Based on the earlier results shown in Figs. 4 and 5, stratospheric air mass as diagnosed solely by PV= 2 PVU is characterized by high $O_3$ mixing ratios around 100 ppbv and lower RH. The scatter-plots in Fig. 8 reveal a wide spread of high PV values at low values of RH and an exponential fit is consequently applied to quantify the

relationships. This exponential fit is performed for four latitude ranges (< 40°N, 40°−50°N, 50°−60°N, and > 60°N) which contain 9, 9, 9, and 6 ozonesonde observational sites, respectively. The results of the exponential fit are summarized in Table 5. Generally, a stratospheric air mass diagnosed by PV = 2.0 PVU is characterized by RH values of 40-50% (Fig. 8(a) and (b) and Table 5). The relation between modeled PV and both modeled and observed RH show a slight dependence on latitude with



lower RH for a given value of PV at lower latitudes (< 40°N) higher RH at higher latitudes (> 60°N). The results also show approximately 5% higher RH values for a given value of PV when using modeled rather than observed RH because of the positive bias of modeled RH. The scatter-plots in Fig. 8 (a) and (b) show large variations with PV especially for higher values of RH. This can be caused by the large variation of PV near the surface due to turbulence as shown in Figs. 4 and 5. To estimate

a more robust relationship between PV and RH, the data from the boundary layer are removed, and the scatter-plots (shown in on Fig. 8 (c) and (d)) are recreated. The results of this new estimate is included in Table 5. Based on these refined results, stratospheric air masses diagnosed by PV = 2.0 PVU are characterized by RH values of 30-40%, and again are approximately 5% higher when using modeled RH rather than observed RH. Since the stratospheric air mass itself can be characterized as a dry air mass (RH with below 10%, Figs. 4 and 5), it is concluded that a stratospheric air mass diagnosed at the tropopause is

characterized by 30-40% RH based on the relation between PV and RH, with lower RH at lower latitudes and higher RH at higher latitudes.

The results presented above are used to develop the air mass characterization method illustrated by the flowchart in Fig. 9. The method relies on modeled $O_3$ and O3PV mixing ratios, RH, and PV. Because the top layer is set to 50 hPa in these H-CMAQ simulations, the uppermost layer (layer number is 44) is always regarded as stratospheric air mass in this method.

For all layers below (i.e., layer 43 down to the lowermost layer), the importance of photochemistry is determined based on the ratio of the O3PV and $O_3$ mixing ratios. As noted in the discussion related to Figs. 4 and 5, if the $O_3$ mixing ratio is higher than the O3PV mixing ratio, it implies that photochemical production affected the air mass, and vice versa. Therefore, a O3PV/$O_3$ ratio of less (more) than 1.0, is classified as photochemical production (destruction), and a value near 1.0 can be classified as weakly impacted by photochemistry. Further, O3PV/$O_3$ ratio less than 0.5 (more than 1.5) is classified as strong photochemical

production (destruction). This classification is then illustrated in Fig. 10, wherein locations and times colored as dark orange (blue) represent air masses influenced by strong photochemical production (destruction), while ratios near 1.0 are colored as light orange (blue). The next step in the classification scheme is to determine whether an air mass is of stratospheric origin. By using the relationship between PV and RH for four latitude ranges as summarized in Table 5, specifically the relationship based on modeled PV and modeled RH excluding the boundary layer data (Fig. 8(c)) an air mass can be judged to be of

stratospheric origin. At least, a PV value of 1.0 PVU and corresponding RH (from Table 5) are required to judge stratospheric origin. To express their strength, a PV value of 1.0 PVU and corresponding RH for four latitude ranges (e.g., 43.6% for < 40°N sites) are considered to be a weak influence, 2.0 PVU and RH to be moderately influenced, while 3.0 PVU and RH are classified as a strong stratospheric intrusion event. Following this determination, further criteria are set to determine a stratospheric intrusion. As discussed above when establishing the PV-RH relationships, near surface air with higher PV due to turbulence

could be diagnosed as stratospheric air mass by only applying PV; therefore, here the PV-RH relationships shown in Fig. 5 and listed in Table 5 are based. Furthermore, the concept of a sequential intrusion from upper layers to lower layers is considered. When the grid cell directly above or surrounding a grid cell are also diagnosed as stratospheric air mass, the grid is determined as being dominated by stratospheric air mass. Applying this concept of a sequential stratospheric air mass intrusion is the final step in the air mass characterization scheme to determine whether an airmass is dominated by




photochemistry or stratospheric intrusion. It is important to note that characterizing a grid cell as being dominated by a process does not mean that other processes do not impact $O_3$ mixing ratios as well. For example, $O_3$ in a grid cell near the tropopause can be dominated by stratospheric air mass, but it can also be affected by photochemical production and destruction. Similarly, although $O_3$ in a grid cell near the surface layer is often dominated by photochemical processes, it can also be affected by stratospheric air mass.

An illustration of applying this method for the six ozonesonde sites in the U.S.A. is presented in Fig. 10. The dominant processes of photochemical production, photochemical destruction, and stratospheric intrusion are colored by warm colors, cool colors, and purple, respectively. In general, the dominant processes can be divided into two parts, with the upper model layers being dominated by stratospheric air and the lower model layers being dominated by photochemical production. At Hilo (Fig. 10 (a)) and Trinidad Head (Fig. 10 (b)), photochemical destruction is dominant below 2 km and 4 km, respectively, as previously illustrated in Fig. 4 by comparing the $O_3$ and O3PV mixing ratios. In Fig. 10, horizontal lines indicating 750, 500, and 250 hPa are also shown. To further examine the impact of stratospheric intrusions on tropospheric $O_3$, time series of daily-mean total $O_3$ tropospheric column mass (gray color) and the inferred tropospheric $O_3$ mass of stratospheric origin calculated from grid cells diagnosed as dominated by stratospheric intrusion (purple color) are shown below the curtain plot. The troposphere is defined as all layers below 250 hPa for the purpose of this analysis. These time series reveal large temporal variations of stratospheric air mass in troposphere. On a monthly-mean basis, air masses classified as being dominated by stratospheric intrusion contribute about 10% to the total tropospheric $O_3$ column mass at five of the ozonesonde sites and 25% at Trinidad Head. However, on specific days, $O_3$ masses from the stratosphere contribute up to 40-70% of the total tropospheric $O_3$ column mass. It should be noted that since the classification scheme is based on the most dominant process, a grid cell classified as being dominated by photochemistry can still be influenced by stratospheric air. Therefore, these estimated impacts of stratospheric air masses on the troposphere can be viewed as a lower bound.

## 3.3 Investigation of Stratospheric Intrusion

The previous section introduces an approach to identify cases when stratospheric air masses impact tropospheric $O_3$. Results of the $O_3$ column mass analysis identify periods in early, middle, and late April 2010 that are affected by stratospheric intrusions over the contiguous U.S.A., and in this section these events are further analyzed. Daily maps of the spatial patterns of the percentage of tropospheric $O_3$ column mass diagnosed as being of stratospheric origin during (i) early and (ii) late April are presented in Fig. 11 while maps for mid-April are presented in Fig. S2 in the supplemental material. On 5 April, a large impact from the stratosphere was seen over the western U.S.A. (indicated as point $S_A$ on the map) and covered Trinidad Head where the contribution of $O_3$ from the stratosphere to the tropospheric $O_3$ mass is 56% (Fig. 10 (b)). This air mass moved eastward on 6 April when the impact at Boulder is 59% (Fig. 10 (c)). During 7-8 April, this meandering air mass was located over the central U.S.A., and then moved further to the east, with a 37% contribution at Huntsville on 9 April (Fig. 10 (d)). Finally, this air mass moved towards the northeast U.S.A. with estimated 39% and 46% stratospheric contributions at Wallops Island (Fig.





10 (e)) and Rhode Island (Fig. 10 (f)), respectively. The stratospheric impacts in early April are associated with an air mass movement from west to east within 5 days; corresponding to an average speed of about 8-9 m/sec. Compared to the early April case, the case in late April is different. On 25 April, large impacts from the stratosphere were found over the southern U.S.A. (indicated as point $S_{B1}$ on the map) and over a stretch from western Canada to northern Colorado (marked as $S_{B2}$). On 26 April,

the $S_{B1}$ air mass affected Huntsville with a stratospheric contribution of 27% (Fig. 10 (d)). While the $S_{B1}$ air mass swept southward on 27 April, the $S_{B2}$ air mass moved south on 26 April and impacted Huntsville with a contribution of 38% on 27 April (Fig. 10 (d)). While the time series from the $O_3$ mass analysis shown in Fig. 10(d) suggest that the stratospheric impacts at Huntsville in late April are from a continuous event, the maps in Fig. 11 reveal that they were caused by different air masses on 26 and 27 April. A third air mass located in the western U.S.A. on 26 April (marked as $S_{B3}$) moved from west to the central

U.S.A., and had large impacts from 27 to 29 April at Trinidad Head with 62% and 61% on 28 April and 29 April (Fig. 10 (b)), affected Boulder on 29 April with a stratospheric contribution of 39% (Fig. 10 (c)), and finally moved westward in a U-shaped pattern on 30 April. Another air mass located over eastern Canada on 26 April (marked as $S_{B4}$) moved slowly southward and impacted Wallops Island (Fig. 10 (e)) and Rhode Island (Fig. 10 (f)) with stratospheric contributions of 44% and 45% on 28 April, then moved eastward. Thus, for the late April case, stratospheric air was present in different air masses impacting

different locations on different days rather than a single air mass simply moving from west to east as in the early April case. Contrasting the early and late April cases illustrates that different synoptic flow scenarios influence how stratospheric air can impact tropospheric $O_3$ columns over the U.S.A.

The impacts of STT during the middle of April are shown in Fig. S2 in the supplemental material. From 12-15 April, tropospheric $O_3$ columns over the western U.S.A. were dominated by stratospheric intrusion, and these impacts shifted to the

20 eastern U.S.A. during 17-19 April. Previous studies (e.g., Lin et al., 2012b) estimated thirteen STT events during April-June 2010, and April 7, 9, 12-15, 21-23, and 28-29 2010 were reported as STT events. Our investigations based on the air mass characterization method matches with these earlier findings. The impact of STT at Huntsville has been investigated by combining ozonesonde and ozone lidar data (Kuang et al., 2012). In their report, the period of 27-29 April was associated with STT, and the estimated $O_3$ mass irreversibly transported into the troposphere was reported to be between 73-106 Gg. The

25 estimation of the $O_3$ air mass of stratospheric origin during this two-day period in this work is 39 Gg out of a total tropospheric $O_3$ mass of 150 Gg. Considering that H-CMAQ tends to underestimate tropospheric $O_3$ mixing ratios (Fig. 5 (d)) and that our air mass characterization method can be viewed as a lower bound of the impacts of stratospheric intrusion, the estimates in this work compare reasonably well with these previous estimates.

Finally, the relationship between the model estimated stratospheric contribution to the total tropospheric $O_3$ column

and observed surface $O_3$ levels at CASTNET sites is investigated in Fig. 12. Using data from all CASTNET locations, the relationship showa a negative slope, indicating that the influence of stratospheric air decreased with increasing surface MD8O3 mixing ratios. To further focus on this relationship at elevated sites in the U.S.A., the analysis is repeated using data from sites with an elevation higher than 1000 m as listed in Table S3 in the supplemental material. These results show an almost flat slope, which indicates that at elevated sites, STT has a near constant effect on surface mixing ratio values that does not change




with the magnitude of surface mixing ratios. The finding of a negative slope using the entire dataset over the U.S.A. is consistent with a previous investigation focused on relatively-polluted areas over the western U.S.A. such as the Central Valley, Southern California, and Las Vegas (Lin et al., 2012b). On the other hand, for elevated sites, they reported a positive slope indicating higher contributions of stratospheric air masses during periods of elevated surface $O_3$. The reason for the difference between this earlier study and this study seems to stem from differences in simulated stratospheric $O_3$ mixing ratios. Lin et al. (2012b) used the fully-coupled stratosphere-troposphere chemistry model GFDL AM3 which tended to overestimate $O_3$ mixing ratios; therefore, they employed a bias correction approach (assuming that when the estimated stratospheric contribution exceeds the model bias, the bias is caused entirely by excessive stratospheric $O_3$) for estimating the stratospheric impacts on surface $O_3$. On the other hand, the H-CMAQ simulations analyzed in this study tends to underestimate tropospheric $O_3$ levels, especially during STT events, which may suggest that its estimates of stratospheric contributions to high surface $O_3$ events may also be too low.

## 4 Conclusions

In this study, the regional chemical transport model CMAQ recently extended for hemispheric applications, the H-CMAQ, is applied to investigate trans-Pacific transport during April 2010. This part 1 manuscript presents results from comprehensive evaluation and a new air mass characterization method based on these results and ozonesonde measurements. The comparison of modeled and observed $O_3$ at the surface shows a very good performance with NMBs around −10%. Comparisons of vertical $O_3$ distributions against ozonesonde and aircraft-based observations show that the model can capture well $O_3$ variations within boundary layer similar to those at the surface, although systematic underestimations of free troposphere $O_3$ occur with NMBs up to −30%, especially during events that are characterized to have strong STT during late April. Modeled RH exhibits a positive bias with NMBs of +10% or greater at all altitudes. Comparisons of modeled tropospheric $O_3$ column with satellite observations suggest that the model has overall good skill in representing the large scale $O_3$ distributions across the Northern Hemisphere with lower column $O_3$ over the Pacific Ocean near equator and higher column in the mid-latitudes.

Using ozonesonde measurements, the relationship between PV and RH is examined to characterize stratospheric air masses. The PV-RH relation indicates that PV of 2.0 PVU (1 PVU = $10^{-6}$ m$^2$ K kg$^{-1}$ s$^{-1}$) generally corresponds to RH values of 30-40%. A new air mass characterization method is further developed based on the ratio of modeled $O_3$ and stratospheric $O_3$ tracer mixing ratios to examine the relative importance of photochemistry, and PV and RH to examine the stratospheric air mass. The estimated STT impacts show significant day-to-day variations both in the magnitude of the contribution and the origin of the air mass. The relationship between surface $O_3$ levels and estimated stratospheric air mass in troposphere exhibits a negative slope, indicating that at most locations, high surface $O_3$ mixing ratios typically result from other factors (e.g., emissions). In contrast, at elevated sites the relationship exhibits a nearly flat slope, indicating a steady STT contribution to $O_3$ levels at these locations.





Despite the use of a coarse horizontal grid resolution for H-CMAQ simulations in this work, it was found that the model performance at the surface and in the boundary layer are reasonably within the model performance criteria suggested from regional-scale applications. However, this work also shows that the model has difficulty capturing higher $O_3$ mixing ratios in the free troposphere through the comparison with ozonesonde data. This result suggests a need for model

improvements to accurately represent the STT process. STT impacts elevated sites in the U.S.A. constantly; however, its impact for greater surface $O_3$ is limited. This study focuses on April 2010, and monthly or seasonal behavior of STT, and interannual variations based on the long-term trend analysis should also be considered for future study. The Part 2 paper will focus on other factors that affect surface $O_3$ mixing ratio, namely emissions.

**Code availability**

Source code for version 5.2 of the CMAQ model can be downloaded from https://github.com/USEPA/CMAQ/tree/5.2. For further information, please visit the US Environmental Protection Agency website for the CMAQ system: https://www.epa.gov/cmaq.

**Data availability**

The observational datasets used in this study are available from their respective websites: http://ds.data.jma.go.jp/gmd/wdcgg/ (WDCGG), http://www.eanet.asia/index.html (EANET), and https://www.epa.gov/castnet (CASTNET) for surface observation network, https://woudc.org/home.php (WOUDC) and https://www.esrl.noaa.gov/gmd/ozwv/ozsondes/ (NOAA ESRL) for ozonesonde, https://www.esrl.noaa.gov/gmd/ozwv/aircraft/index.html (NOAA-ESRL) for airplane, https://acd-
20 ext.gsfc.nasa.gov/Data_services/cloud_slice/index.html (NASA). Last Access: 31 August 2018.

**Competing interests**

The authors declare that they have no conflict of interest.



**Disclaimer**

The views expressed in this paper are those of the authors and do not necessarily reflects the views or policies of the U.S. Environmental Protection Agency.

**Author contributions**

Syuichi Itahashi performed the analysis of observation and model simulation and prepared the manuscript with contributions from all co-authors. Rohit Mathur and Christian Hogrefe contributed to establish the hemispheric modeling application for this study and prepared the emission dataset and initial condition from previous long-term simulation results. Yang Zhang contributed to the literature review of trans-Pacific transport and refined this research through simulation designs, and results
interpretation.

**Acknowledgement**

The authors are grateful the observation dataset of surface (WDCGG, EANET, and CASTNET), ozonesonde (WOUDC and NOAA ESRL), airplane (NOAA ESRL), and satellite (NASA). YZ acknowledges support from the 2017-2018 NC State
Internationalization Seed Grant and the 2019-2020 NC State Kelly Memorial Fund for US-Japan Scientific Cooperation.

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





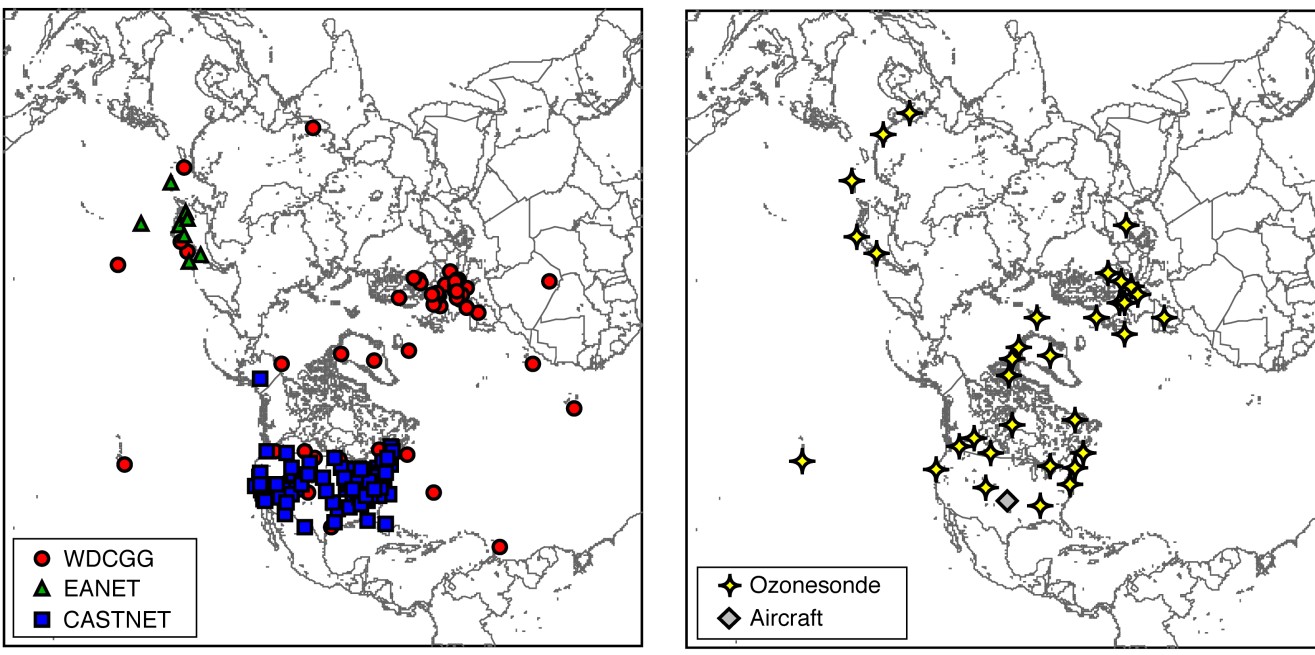

**Figure 1. Geographical mapping of the (left) surface and (right) aloft observational sites used in this study. Detailed information about the ozonesonde observation sites is provided in Table 1.**





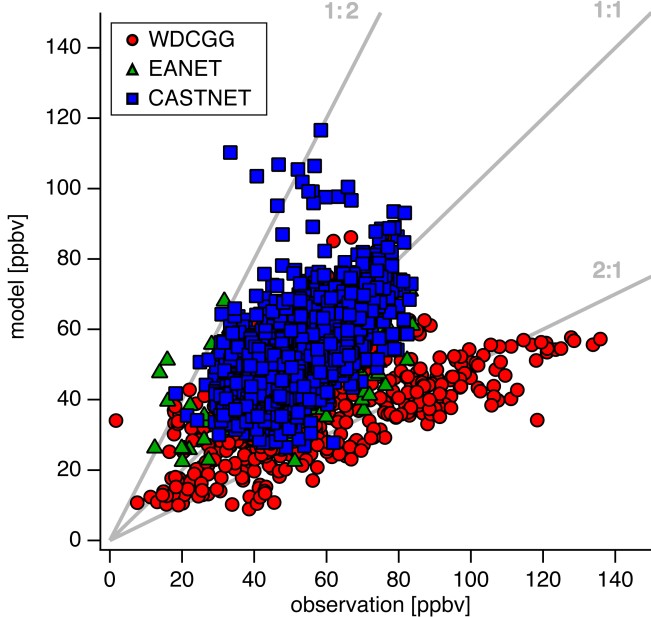

**Figure 2. Scatter-plot between observations and H-CMAQ simulations for surface MD8O3 during April 2010. Reference lines are provided at ratios of 2:1, 1:1 and 1:2. The symbols and colors used to represent the different surface observational datasets are consistent with Fig. 1.**



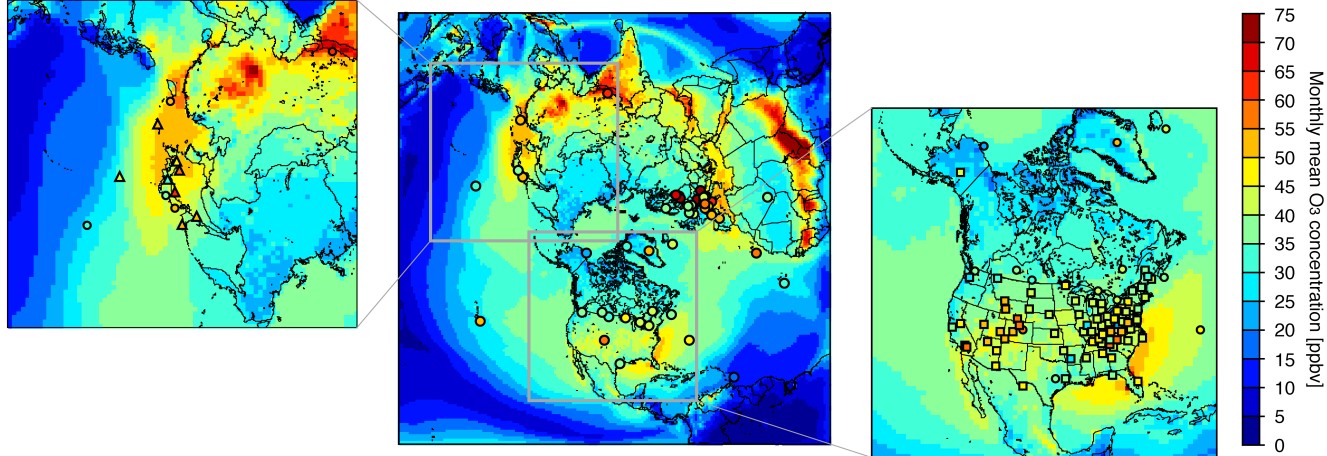

**Figure 3. Monthly-mean H-CMAQ O₃ mixing ratios overlaid with WDCGG surface observations, and zoom-in panels over (left) Asia overlaid with EANET surface observations, and (right) U.S. overlaid with CASTNET surface observations.**





**Figure 4.** Curtain plots of modeled (left) O₃, (center) O3PV, and (right) RH at U.S. ozonesonde sites of (a) Hilo (HI), (b) Trinidad Head (CA), and (c) Boulder (CO) during April 2010. Yellow stars indicate the time of available ozonesonde measurements. Contour lines of modeled PV are also inserted for contours of 1.0, 1.5, 2.0, 2.5, and 3.0 PVU with thick lines denoting the 2.0 PVU contour as an index to diagnose the tropopause.




**Figure 4. Continued, but at (d) Huntsville (AL), (e) Wallops Island (VA), and (f) Rhode Island (RI).**





(a) Hilo

(b) Trinidad Head

(c) Boulder



**Figure 5. Vertical profiles of observed and modeled O₃ and RH at U.S. ozonesonde sites of (a) Hilo (HI), (b) Trinidad Head (CA), (c) Boulder (CO). Also see Figure 4 for ozonesonde measurement times. For modeled O₃ and RH, the hourly result corresponding to the ozonesonde measurement time is shown by circles, and the maximum and minimum model results within ±2 hours of the measurement time are shown by whiskers. For observed O₃ at Hilo and Boulder, the range of uncertainties of the O₃ observations is shown by whiskers. Modeled O3PV and PV are also shown. Modeled PV profiles are plotted in red, and vertical lines corresponding to a PV value of 2 PVU are inserted as an index of the tropopause, and the layer range diagnosed as stratospheric air mass is colored in purple.**





**(d) Huntsville**



**(e) Wallops Island**

**(f) Rhode Island**

**Figure 5.** Continued, but at (d) Huntsville (AL), (e) Wallops Island (VA), and (f) Rhode Island (RI).





**Figure 6. Curtain plot and vertical profiles of O₃ and RH at the aircraft observational site: Southern Great Plains, Oklahoma. Gray diamonds indicate the times of the seven aircraft flights, and results for these flights are shown in the expanded boxes for 5-h time windows overlaid with observations. Observed ascent and descent profile data are averaged into 100-m grid resolution, and profiles of the mean and standard deviations are shown. Modeled ascent and descent data corresponding to the observation times are averaged on original modeled layers, and the mean and standard deviations are shown.**



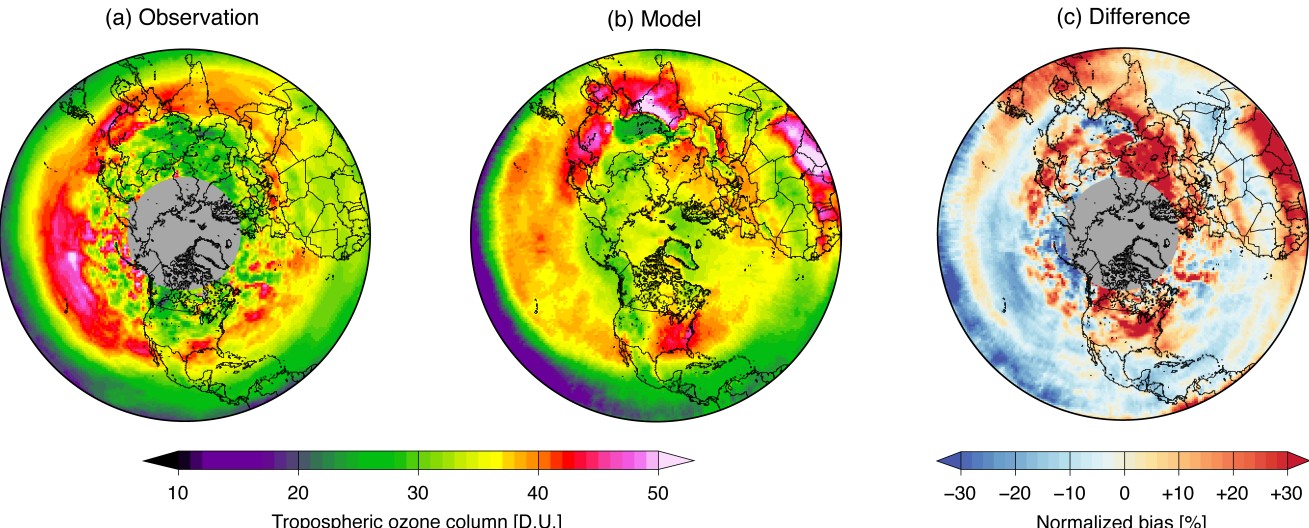

**Figure 7. Tropospheric ozone column of (a) satellite observation, (b) H-CMAQ simulation, and (c) their differences shown as normalized bias. Areas filled in gray colored indicate missing observations.**



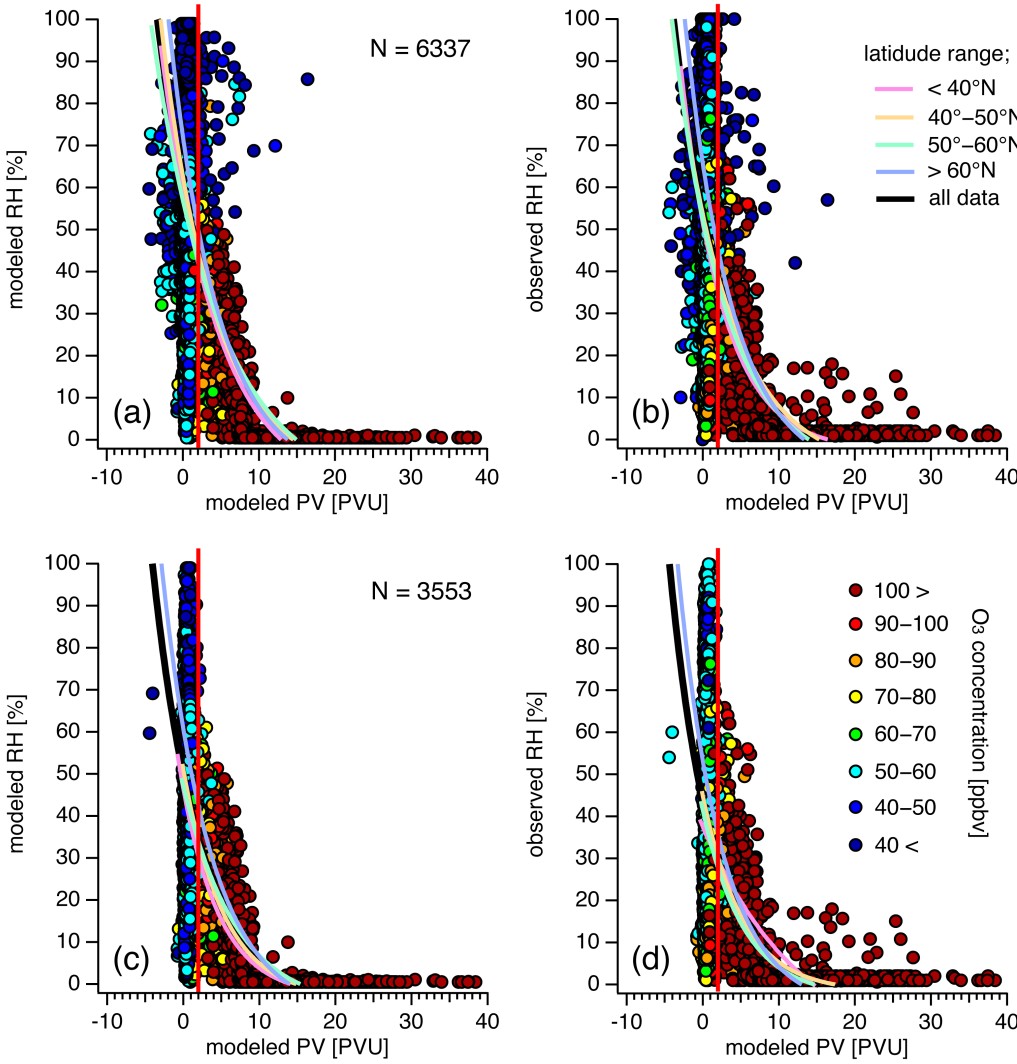

**Figure 8.** PV-RH relationship during April 2010 for (a) modeled PV and modeled RH using all data, (b) same as (a) but using observed RH, (c) same as (a) but removing data within the boundary layer, and (d) same as (c) but using observed RH. The points in these scatter-plots are color-coded based on the (a, c) modeled and (b, d) observed $O_3$ mixing ratio range corresponding to each PV-RH data point. Exponential fits over four different latitude bands (<40°N, 40-50°N, 50-60°N, >60°N) are presented by lines of different colors, and by a thick black line when using all available data for the fit. Also see Table 5 for the results of the exponential fitting.





**Figure 9. Flowchart of the air mass characterization scheme. See the text for additional details.**



**Figure 10. (Top)** Curtin plot of model-diagnosed air mass characterization and **(bottom)** daily O₃ mass in the troposphere (gray color) and stratospheric air mass in the troposphere (purple color) at U.S. ozonesonde sites of **(a)** Hilo (HI), **(b)** Trinidad Head (CA), **(c)** Boulder (CO), **(d)** Huntsville (AL), **(e)** Wallops Island (VA), and **(f)** Rhode Island (RI) during April 2010.



**Figure 11. Spatial distributions of day-to-day variations of stratospheric air mass contributions to total tropospheric $O_3$ column mass over the U.S. during (i) early, and (ii) late April 2010. Yellow stars indicate the ozonesonde observational sites. Red character strings ($S_A$ and $S_{B1}$-$S_{B4}$) denote the different stratospheric air masses discussed in the text.**





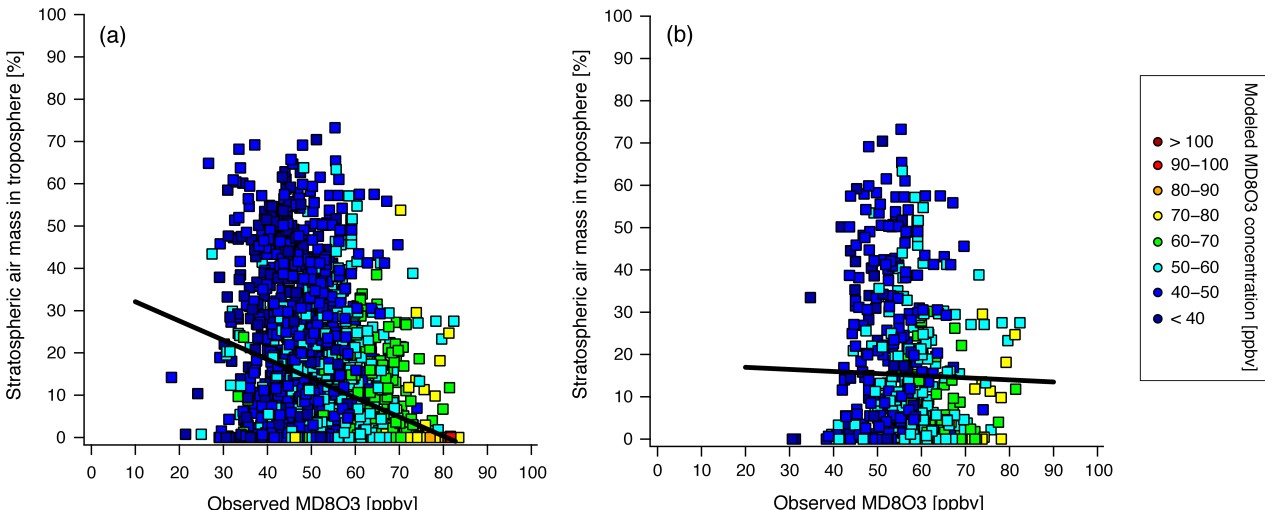

**Figure 12.** Relationship between observed MD8O3 at the surface and the estimated stratospheric air mass contributed to total tropospheric O₃ column mass. The points are color coded based on modeled MD8O3 mixing ratios. (a) All CASTNET sites, and (b) elevated CASTNET sites defined as having an elevation greater than 1000 m (see also Table S3).



**Table 1. Literature review of previous studies on estimated impacts of (upper) emissions from Asia, and (lower) stratospheric intrusion.**

| Time Period | Estimated impacts | Location | Method | Reference |
|---|---|---|---|---|
| April 2001 | 1 ppbv (monthly mean), up to 2.5 ppbv (daily mean) | western U.S. | model (process analysis) | Wang et al. (2009) |
| April-May 2002 | 10 ppbv increase from April-May 1984 | 5 west coast U.S. sites | observation (linear regression) | Jaffe et al. (2003) |
| April-May 2006 | 5-7 ppbv (17 April-15 May 2006; INTEX-B), increased by 1-2 ppbv from April-May 2000 | western North America | model (zero-out, and emission scenario for Asia) | Zhang et al. (2008) |
|  | 2-5 ppbv (17 April-15 May 2006; INTEX-B) | eastern North America | model (zero-out) |  |
| April-June 2010 | 2-6 ppbv increase from April-June 1985 | western U.S. | model (tripling Asian emissions from 1985) | Jacob et al. (1999) |
|  | 1-3 ppbv increase from April-June 1985 | eastern U.S. | model (tripling emissions from 1985) |  |
| May-June 2010 | 8-15 ppbv on specific events | high-elevation regions | model (zero-out) | Lin et al. (2012a) |
| April-June 2010 | 4.7±2.4 ppbv (three-month mean) | 15 high-elevation western U.S. sites | model (zero-out) | Lin et al. (2012b) |
| March-October 2001 | below 20 ppbv | U.S. | tagged $O_3$ | Fiore et al. (2003) |
| April-May 1990-2012 | ranged 10-25 ppbv | 22 high-elevation western U.S. sites | tropopause tracer | Lin et al. (2015) |
| April-June 2010 | 22.3±11.5 ppbv (three-month mean) | 15 high-elevation western U.S. sites | tropopause tracer | Lin et al. (2012b) |



**Table 2. Details of the ozonesonde dataset used in this study.**

| Country | Site name | Longitude (°) | Latitude (°) | Elevation (m a.s.l.) | Data source | # of launch |
|---|---|---|---|---|---|---|
| USA (HI) | Hilo | −155.05 | 19.72 | 10 | NOAA ESRL | 3 |
| USA (CA) | Trinidad Head | −124.15 | 41.06 | 36 | NOAA ESRL | 5 |
| USA (CO) | Boulder | −105.20 | 39.95 | 1743 | NOAA ESRL | 4 |
| USA (AL) | Huntsville | −86.65 | 34.73 | 203 | NOAA ESRL | 5 |
| USA (VA) | Wallops Island | −75.47 | 37.93 | 13 | WOUDC | 6 |
| USA (RI) | Rhode Island | −71.42 | 41.49 | 21 | NOAA ESRL | 2 |
| Canada (BC) | Kelonwa | -119.4 | 49.94 | 456 | WOUDC | 4 |
| Canada (AB) | Edmonton | −114.1 | 53.54 | 766 | WOUDC | 4 |
| Canada (SK) | Bratt's Lake | -104.7 | 50.2 | 580 | WOUDC | 4 |
| Canada (NU) | Resolute | -94.97 | 74.71 | 46 | WOUDC | 2 |
| Canada (MB) | Churchill | -94.07 | 58.74 | 30 | WOUDC | 4 |
| Canada (NU) | Eureka | -85.94 | 79.98 | 10 | WOUDC | 3 |
| Canada (ON) | Egbert | -79.78 | 44.23 | 252 | WOUDC | 4 |
| Canada (NS) | Yarmouth | -66.11 | 43.87 | 9 | WOUDC | 3 |
| Canada (NU) | Alert | -62.34 | 82.49 | 75 | WOUDC | 3 |
| Canada (NL) | Goose Bay | -60.36 | 53.3 | 36 | WOUDC | 3 |
| Vietnam | Hanoi | 105.8 | 21.01 | 7 | WOUDC | 1 |
| China | Hong Kong | 114.17 | 22.31 | 66 | WOUDC | 4 |
| Japan | Naha | 127.69 | 26.21 | 28 | WOUDC | 3 |
| Japan | Tateno | 140.13 | 36.06 | 31 | WOUDC | 3 |
| Japan | Sapporo | 141.33 | 43.06 | 26 | WOUDC | 3 |
| Greenland | Summit | −38.46 | 72.58 | 3211 | NOAA ESRL | 4 |
| Ireland | Valentia | -10.25 | 51.93 | 14 | WOUDC | 3 |
| Spain | Madrid | -3.58 | 40.47 | 631 | WOUDC | 3 |
| UK | Lerwick | -1.19 | 60.14 | 80 | WOUDC | 5 |
| Belgium | Uccle | 4.35 | 50.8 | 100 | WOUDC | 12 |
| Netherland | De Bilt | 5.18 | 52.1 | 4 | WOUDC | 8 |
| Switherland | Payerne | 6.57 | 46.49 | 491 | WOUDC | 12 |
| German | Hohenpeissenberg | 11.0 | 47.8 | 976 | WOUDC | 18 |
| Norway | Ny Alesund | 11.95 | 78.93 | 11 | WOUDC | 6 |
| Czech | Praha | 14.44 | 50.0 | 304 | WOUDC | 10 |
| Poland | Legionowo | 20.97 | 52.4 | 96 | WOUDC | 4 |
| Turkey | Ankara | 32.86 | 39.97 | 890 | WOUDC | 2 |

Note: Parenthesis after the country name indicates the state.



**Table 3. Statistical analysis of modeled O$_3$ concentration using surface, ozonesonde, aircraft, and satellite observations.**

| | N | Mean | | R | NMB | NME |
|---|---|---|---|---|---|---|
| | | Observation | Model | | | |
| Surface | | | | | | |
| −WDCGG | 1498 | 53.9 | 43.5 | 0.49*** | −19.3% | 23.7% |
| −CASTNET | 2316 | 53.4 | 52.9 | 0.61*** | −0.9% | 12.6% |
| −EANET | 240 | 56.2 | 49.1 | 0.49*** | −12.6% | 20.6% |
| Ozonesonde | | | | | | |
| USA and Canada | | | | | | |
| −boundary layer | 1016 | 46.0 | 42.7 | 0.70*** | −7.1% | 16.7% |
| −free troposphere | 893 | 87.7 | 59.6 | 0.79*** | −32.1% | 33.8% |
| −upper model layer | 512 | 905.1 | 770.3 | 0.91*** | −14.9% | 30.2% |
| Asia | | | | | | |
| −boundary layer | 283 | 44.2 | 47.4 | 0.44*** | 7.1% | 24.5% |
| −free troposphere | 207 | 70.7 | 59.0 | 0.43*** | −16.5% | 21.5% |
| −upper model layer | 124 | 529.8 | 399.2 | 0.94*** | −24.6% | 34.4% |
| Europe | | | | | | |
| −boundary layer | 1478 | 47.6 | 46.8 | 0.42*** | −1.6% | 17.9% |
| −free troposphere | 1368 | 78.2 | 57.2 | 0.76*** | −26.8% | 29.1% |
| −upper model layer | 817 | 1015.7 | 894.3 | 0.94*** | −11.9% | 24.2% |
| Aircraft | | | | | | |
| −from sufrace upto 6 km | 128 | 55.9 | 45.3 | 0.74*** | −19.0% | 19.1% |
| Satellite | | | | | | |
| −Tropospheric column | 28020 | 33.2 | 34.7 | 0.65*** | 4.7% | 13.5% |

Note: The unit of mean for observations and simulations is ppbv except satellite observation expressed as D.U. Maximum daily 8 hour average ozone (MD8O3) is used for surface observational data of WDCGG, CASTNET, and EANET. Corresponded hourly modeled O$_3$ is used for ozonesonde data. 2-4 hours averaged hourly modeled O$_3$ is used for aircraft data to fully cover each observation time. Significance levels by Students' t-test for correlation coefficients between observations and simulations are remarked as *p < 0.05, **p < 0.01, and ***p < 0.001, and lack of a mark indicates no significance.



**Table 4. Statistical analysis of modeled RH using ozonesonde and aircraft observations.**

| | N | Mean | | R | NMB | NME |
|---|---|---|---|---|---|---|
| | | Observation | Model | | | |
| Ozonesonde | | | | | | |
| USA and Canada | | | | | | |
| −boundary layer | 1016 | 57.70 | 67.07 | 0.73*** | 16.2% | 24.5% |
| −free troposphere | 881 | 39.16 | 43.31 | 0.83*** | 10.8% | 29.7% |
| −upper model layer | 398 | 7.81 | 8.72 | 0.79*** | 11.6% | 62.3% |
| Asia | | | | | | |
| −boundary layer | 283 | 65.89 | 79.63 | 0.45*** | 20.8% | 28.7% |
| −free troposphere | 184 | 46.26 | 51.52 | 0.38*** | 11.4% | 47.7% |
| −upper model layer | 43 | 18.96 | 26.47 | 0.63*** | 39.6% | 67.2% |
| Europe | | | | | | |
| −boundary layer | 1485 | 63.84 | 68.92 | 0.73*** | 8.0% | 17.1% |
| −free troposphere | 1368 | 36.14 | 42.82 | 0.80*** | 18.5% | 32.6% |
| −upper model layer | 679 | 7.13 | 9.56 | 0.91*** | 34.1% | 56.1% |
| Aircraft | | | | | | |
| −troposphere | 126 | 41.66 | 52.04 | 0.84*** | 24.9% | 28.7% |

Note: Significance levels by Students' t-test for correlation coefficients between observations and simulations are remarked as
*p < 0.05, **p < 0.01, and ***p < 0.001, and lack of a mark indicates no significance. 5-hour averaged hourly modeled relative
humidity is used for ozonesonde data. 2-4 hours averaged hourly modeled relative humidity is used for aircraft data to fully
5 cover each observation time, and original aircraft data are averaged into 100-m resolution to be compared with model.



**Table 5. The relationship between PV and RH based on the comparison at ozonesonde observational sites.**

| PV | modeled RH | | | | | observed RH | | | | |
|---|---|---|---|---|---|---|---|---|---|---|
| | 1.0 | 1.5 | 2.0 | 2.5 | 3.0 | 1.0 | 1.5 | 2.0 | 2.5 | 3.0 |
| all dataset | | | | | | | | | | |
| < 40°N sites | 50.1 | 46.0 | 42.2 | 38.6 | 35.3 | 42.7 | 38.8 | 35.2 | 31.9 | 29.0 |
| 40°−50°N sites | 53.2 | 48.9 | 44.9 | 41.2 | 37.7 | 45.8 | 42.2 | 38.9 | 35.8 | 32.9 |
| 50°−60°N sites | 49.6 | 46.2 | 42.9 | 39.9 | 37.0 | 44.1 | 40.5 | 37.1 | 34.0 | 31.0 |
| > 60°N sites | 60.5 | 55.3 | 50.5 | 46.0 | 41.8 | 54.7 | 49.9 | 45.3 | 41.2 | 37.3 |
| all sites | 52.3 | 48.3 | 44.6 | 41.2 | 37.9 | 45.7 | 42.0 | 38.9 | 35.3 | 32.4 |
| exclude boundary layer dataset | | | | | | | | | | |
| < 40°N sites | 43.7 | 39.8 | 36.3 | 33.2 | 30.3 | 33.4 | 32.2 | 31.0 | 29.9 | 28.7 |
| 40°−50°N sites | 41.8 | 37.8 | 34.1 | 30.7 | 27.6 | 37.0 | 33.5 | 30.3 | 27.4 | 24.8 |
| 50°−60°N sites | 40.1 | 36.9 | 34.0 | 31.2 | 28.6 | 34.0 | 30.7 | 27.7 | 25.0 | 22.5 |
| > 60°N sites | 50.9 | 46.4 | 42.3 | 38.5 | 35.0 | 43.8 | 39.6 | 35.8 | 32.3 | 29.0 |
| all sites | 41.8 | 38.2 | 34.8 | 31.8 | 28.9 | 36.0 | 32.6 | 29.5 | 26.7 | 24.1 |

Note: There are 9, 9, 9, and 6 ozonesonde observational sites located in the latitude ranges of <40°N, 40°−50°N, 50°−60°N, >60°N, respectively.