# Peer review of "Modeling Stratospheric Intrusion and Trans-Pacific Transport on Tropospheric Ozone using Hemispheric CMAQ during April 2010: Part 1. Model Evaluation and Air Mass Characterization for Stratosphere-Troposphere Transport"

_Atmospheric Chemistry and Physics, 2019_

## Referee Comment (RC1) · Anonymous Referee #1 · 3 Jun 2019

Itahashi et al. (2019) investigated the impacts of stratospheric intrusion on tropospheric ozone based on the relationship between potential vorticity and relative humidity. They found high surface O3 are often associated with emissions whereas stratospheric intrusion contribute to O3 at elevated sites. The manuscript is in general well written. Below are a few comments need to be addressed.

General comments:

[Figure]

Tropopause height

In this work, tropopause is determined at 2.0 PVU. How is the model performance in simulating PV? How different the tropopause height calculated in this work from the traditional approach (e.g., WMO 1992).

Tropospheric O3

O3 is underestimated in the free troposphere in the model. Does the model include lightning NOx emissions? If so, are they prescribed or on-line calculated? Underestimations in lightning NOx emissions could lead to the underestimations in O3.

Trans-pacific transport

Trans-pacific transport is not really discussed in this paper although it is shown in the title. When O3PV/O3 is used to characterize air masses, how do you distinguish air masses from trans-pacific transport?

Specific comments:

Figure 5, this is very complicated figure and includes a lot of information. Is there any way to evaluate PV?

Regarding O3PV and O3, should they be overlapping in stratosphere that you defined based on 2PVU? Note there are some differences between these two (e.g., at Huntsville site). Any explanations on that?

For observed RH profiles, in most cases, there is a steep decrease in RH from tropopause to upper layers. But at Wallops Island site, there is no such large decrease in RH, especially in early to middle April while the model shows a decreasing trend. Any explanations?

Figure 6, the profiles (row 5) are too small. On page 11, line 5, "flight #6 might be a case of STT because observed RH is less than 10% and observed O3 mixing ratios exceed 75 ppb", where is the tropopause for this case, below or above 6km?

Figure 9, how do you distinguish the impacts of horizontal transport and stratospheric intrusion?

Other comments

There are a few places with grammar errors.

Page 3, line 13-16, "On one hand. . ., on the other hand. . ..", split into two sentences.

Page 10, line 26, "over 500 ppbv at around 8 km The profiles . . ." these are two sentences.

Page 11, line 16, "Europe, the, model . . ." need correct

Page 12, line 1, " lower RH . . .at lower latitudes (<40N) higher RH at higher latitude. . ." need correct grammar error

Page 12, line 30-31, ". . .listed in Table 5 are based.", based on what? Incomplete

---

## Referee Comment (RC2) · Anonymous Referee #3 · 11 Jun 2019

Review of the paper

Modeling Trans-Pacific Transport using Hemispheric CMAQ during April 2010: Part 1. Model Evaluation and Air Mass Characterization for the Estimation of Stratospheric Intrusion on Tropospheric Ozone by Syuichi Itahashi and co-authors

The paper is the first in a series of two investigating CMAQ simulations of trans-pacific pollution transport for one month (April) in 2010. In contrast to what the main title

promises the paper deals exclusively with ozone transport from the stratosphere and a method for a quantification of the stratospheric contribution to tropospheric ozone. The approach followed here to answer this question is quite interesting and generally well described. However, the study suffers from the extremely short period under investigation. The authors justify why they selected this period for their study. They claim that it was published earlier that trans-Pacific transport played an important role during this period (Uno et al., 2011, Lin et al., 2012a). However, they try to draw more general conclusions about the contribution of stratospheric ozone to the concentrations in the troposphere and the model performance. I believe these findings on biases and model skill are not well justified because the data set used for this type of evaluation is too small. I suggest that the authors investigate a longer simulation period in order to derive statistical parameters for the model performance and the stratospheric contribution to tropospheric O3. They can then still investigate April 2010 as a special case in more detail like it is done now.

Specific major comments:

Page (P) 1, line (l) 21: You describe the bias in RH given by H-CMAQ, however RH should be a quantity given by the driving meteorology model, which in this case is WRF.

P 1, l 28-32: These statements are based on an investigation for April 2010 but the reader gets the impression that they have a more general validity. You should extend your evaluation period for this type of conclusions.

P 2, l 17: Is it still true that the emissions in East Asia increase? There are more recent publications reporting the contrary.

P 3, l 1/2: How can a publication from 1999 say something on real trends until 2010? You need to point out that this was a model study looking into the future. In addition, it would be interesting to know if the predictions for the development of Asian emissions were correct.

[Figure]

P 3, l 8/9: When Lin et al. modelled May/June 2010 and you do April 2010: Why don't you extend your model period and put the results in perspective to their results?

P 3, l 19-21: Because this is the case you need to cover other seasons with your model in order to evaluate it properly.

P 3, l 29/30: This objective is not covered in this paper at all. You should say more clearly what the objective of this paper is.

P 5, l 1 / 2: You state that O3 in the stratosphere is parameterized based on PV from WRF and an O3-PV function from Xing et al. (2016). Could you say a few words about how accurate this parameterization is?

P 5, l 17-19: Why did you simulate only such a short period? Is it computationally expensive to run the model? Which boundary conditions of those reported in the Hogrefe et al. (2018) paper did you use?

P 8, l 13-30: Again, given these significant deviations between model results and observations, it would be beneficial for your interpretation if you extend the modelled time period.

P 10, l 8/10: Doesn't this suggest that the scaling approach is not accurate enough for modelling ozone concentrations in the upper troposphere. So isn't there a need for adding a model component that covers stratospheric ozone with its entire chemistry and dynamics?

P 10, l 19-21: What could be the reason for this positive bias if it occurs despite nudging of RH from reanalysis data?

P11, l 10-20: The comparison of the ozone profiles to the model values (Table 3) suggest that the model gives too low O3 concentrations, in particular in the free troposphere. You also state this in l 9/10 on P 11. However, in Figure 7 we see a mostly positive bias with too high modeled column values, in particular over the continents, where the O3 soundings were performed. Could you explain this? Does it tell me that

satellite observations deviate quite much from ozone soundings? And oes it mean that your findings whether the model is too high or too low depends on the observations you compare it with?

P 13, l 16-18: This is another example for the main problem of this study: You investigated April 2010, only, but you give the impression that you could draw more general conclusions out of it. You should extend the modeled time series in order to give these conclusions.

P 15, l 21-24: Which measure did you use for saying the model has good skill for representing the main hemispheric O3 distribution? The model is obviously too high over Africa in the equator region and it shows higher values over continents and lower values over oceans compared to the satellite observations.

Minor comments:

P 2, l 16: do you mean that the number of low O3 days increased or the concentrations on the low O3 days? These are very different things and it is not clear, here.

P 14, l 31: shows

P 15, l 31/32 and P 16, l 8: which emissions lead to high surface O3 mixing ratios?

P 17 – 22: The references need to be revised with respect to formatting and initials.

Figures:

Figure 2: It is impossible to judge the distribution of the red and the green points when the blue squares are plotted in this way.

Figure 4 and Figure 5: These plots look nice but I think not all of them are needed. You may put some of them into the supplemental material.

Figure 8: It is not clear to me what the exponential fit stands for. Is it used somewhere else?

[Figure]

---

## Referee Comment (RC3) · Anonymous Referee #2 · 12 Jun 2019

This manuscript is the first part of at least two parts of a paper series dedicated to the analysis of trans-Pacific transport. This first part is focused on the evaluation of the WRF / H-CMAQ model configuration and on the analysis of stratospheric intrusion. The thorough analysis in the manuscripts has two flaws:

First, the model simulation uses a horizontal grid spacing of 108 km, which is a very coarse resolution to realistically simulate stratospheric intrusions.

Second, unfortunately, the authors are omitting vital information about their most

important diagnostic tool, the O3PV tracer. On the definition provided in the article the diagnostic method described in Sect. 3.2 seem to be not fully applicable and thus I doubt the results of Sect. 3.3.

Therefore, depending on the real definition (in contrast to my unterstanding of the description in the manuscript) of O3PV I am rating the manuscript as either reject or major revisions.

**Major Issues**

- p. 4 l. 15-20: Looking at the very coarse horizontal resolution of 108 km, it might be nice, that the 44 layer version represents STT better than the 35 layer version. However, the horizontal resolution is much too coarse to expect a good representation of the downward mixing during STT events. (e.g., Gray 2003, Cristofanelli et al., 2003).
  This alone compromises the usefulness of this study.

- p. 4 l. 25: "The value of PV generally increases with altitude ...": depending on the shape of the stratospheric intrusion / the PV streamer this is precisely not necessarily the case.

- p. 4 l. 30 - p. 5 l. 4: The definition of the O3PV tracer is not clear. How is this tracer initialised? When (at initialisation, each step ...) and where (free tropopause, stratosphere ...) is this O3-PV relationship used to define the O3PV tracer and how (is O3PV set to O3 in respective regions)? All this is essential for the information this tracer is carrying, thus a much more detailed explanation is required here.

- p. 7 l. 2/3 What about high-PV structures in the free troposphere? Are they simply declared to be stratospheric?

- p. 9 l. 2-4: "Generally, O3 and O3PV mixing ratios are very similar in the upper layers, especially above the 2.0 PVU line, indicating that O3 mixing ratio in these layers are dominated by stratospheric air mass. "
  I thought that is the definition of the O3PV tracer, how could these tracers not be very similar?

- p. 10 l. 9/10: more importantly the horizontal resolution needs to be increased.

- p. 10 l. 11-21: What do you expect? RH is a diagnostic quantity which is dependent on a bundle of prognostic variables and sensitive parametrisations. Thus RH is a very difficile variable to base further analysis on.

- p. 10 l. 22-30: You show here that RH is far from realistic in the model but still the new analysis method in 3.2 is based on this diagnosed quantity?

- p. 11 l. 10-20 / Fig. 7: I can not agree, that the model captures the observation well. The only thing that is correct is the location of the maximum over the Pacific Ocean.

- Sect. 3.2

  – p. 11/12 / Fig. 8 / Table 5: From the data provided here, I can not agree to the method how the relationship between PV and RH is established. There is no proof, that the exponential fit is the best one. Table 5 does not provide any statistical measures to assess the quality of this fit. Maybe an elephant might have been an option too?

  – I can think of low humidity conditions without stratospheric influence (e.g. above deserts).

  – p. 12, ll. 9ff.: How do you deal with high-PV structures in the troposphere. Where is the tropopause diagnosted in these cases?

– p. 12/13: too understand this method it is essential to understand how the O3PV tracer is initialised. As explained above, the description provided in this manuscript is not self-explanatory. I assume: the O3PV tracer is set every time step to O3 where PV is higher than 2 PVU (this might include high-PV structures in in the troposphere) and might blur the signal of "real" stratospheric air.:

– Additionally, as the O3PV tracer is transported and depositioned due to its own gradients many deviations between the Ozone and the O3PV tracer might be caused by differences in transport and sinks and not in photo-chemistry.

– Fig. 10 and corresponding text: The description of you results reads as if stratospheric ozone would be inert and only tropospheric ozone would take place in photochemistry. The fastest process of all are the autocat-alytic cycles of ozone production and destruction. Therefore, the amount of stratospheric ozone influences directly the photochemisty. How is this stratospheric ozone mass calculated? Is it the integral over O3PV? In that case, I would say that the assessment is wrong as you miss its photochem-ical sink. (provide more details about the calculation p.13, ll.12-14)

– From the current knowledge about the method I would say, that a continu-ously initialised stratospheric tracer could be a diagnostic tool to diagnose stratospheric influence. But the quantification diagnostic introduced in Sect. 3.2. does not work, unless the authors omitted to provide a lot of vital infor-mation about their method.

• Sect. 3.3: As I question the diagnostic method explained in Sect. 3.2, I have to doubt the results of this section as well. Of course you can say, whether the air is influenced by stratospheric air, but the percentages provided in Fig. 11 mean nothing.

[Figure]

- p. 15 l. 30-33: Due to the coarse horizontal resolution of the model it was not to be expected that stratospheric ozone is transported downward efficiently enough to reach the surface.

**Minor Issues**

- title: should contain the model version, as evaluations are always specific for the used model version. Additionally, the title is misleading as the authors miss to point out the interdependencies between STT and trans-Pacific transport.

- p. 1 l. 29/30: not clear what the message is. Where else could STT impacts come from?

- p. 2 l. 1: as STT is event based I doubt that the impact is near constant.

- p. 2 l. 17: "acceleration of anthropogenic emissions" ? emissions are not accelerated. They might increase and their increase might be accelerated ...

- p. 4 l. 21: What is cb05e51? A GIT tag ?

- unify "O3/PV" vs. "O3-PV" relationship.

- Sect. 2.1: Are these (WRF and H-CMAQ) continuous simulations or are they re-initialised?

- Sect. 2.2.3 It is really necessary to talk about un-used flight data?

- Fig. 1: Usage of lighter colors would make it easier to see the symbols. The grey aircraft symbol is hard to distinguish from the grey map lines.

- longitude / latitude information is missing in all maps

- Fig. 4: thick line not identifyable, black lines are distinguishable only at 300 % zoom and more.

- Table 1: The tables content is not understandable without providing more details, e.g.:

  – What does "ranged" and "zero-out" mean?
  – "tagged O3": which tagging method?
  – "tropopause tracer": How defined / initialised?
  – Table 1: the descriptions of the "Estimated impacts" are completely messed, e.g., "5-7 ppbv (17 April- 15 May 2006; INTEX-B), increased by 1-2 ppbv from April-May 2000" What does this mean? The estimate stems from a measurement in 2006 during the INTEX-B compaign and is compared to a 2000 value, where we do not know anything about? And do you mean that it impact increased by 5-7ppbv?

Literature:

- Cristofanelli, P., Bonasoni, P., Collins, W., Feichter, J., Forster, C., James, P., Kentarchos, A., Kubik, P., Land, C., Meloen, J., Roelofs, G., Siegmund, P., Sprenger, M., Schnabel, C., Stohl, A., Tobler, L., Tositti, L., Trickl, T., and Zanis, P.: Stratosphere-to-troposphere transport: A model and method evaluation, J. Geophys. Res., 108, 8525, doi:10.1029/2002JD002600, 2003.

- Gray, S.: A case study of stratosphere to troposphere transport: The role of convective transport and the sensitivity to model resolution, J. Geophys. Res., 108, 4590, doi:10.1029/2002JD003317, 2003.

---

## Short Comment (SC1) · 19 Jun 2019

I share the concerns raised by the 2nd reviewer about model resolution and the description of the setup of the stratospheric tracer O3PV. In addition, I would like to comment on the issue of using PV to identify "air of stratospheric origin". According to Fig. 10, an air mass is classified as a strong/moderate/weak stratospheric intrusion if PV is larger than 3/2/1 pvu. Here it is not fully clear what "strong" and "moderate" mean: if air with

PV>2 pvu is stratospheric then why calling PV>3 pvu "strong intrusion"? It has higher PV but this paper is about ozone. This measure of "strength" does not necessarily reflect the influence on ozone. Things become more problematic in the text where it reads: "At least, a PV value of 1.0 PVU and corresponding RH (from Table 5) are required to judge stratospheric origin." Note that "stratospheric origin" is not the same as a "stratospheric intrusion". An originally stratospheric air parcel that experiences STT, first has high PV (> 2pvu) and, most likely, high O3. It then loses PV due to some diabatic process (turbulence, radiation, convection) and enters the troposphere (what we call STT). Ozone values might still be high. Entering further into the lower troposphere many things can happen: diabatic processes can further reduce PV (to very low values of less than 1 pvu), the air parcel may become moister (due to turbulent mixing) and its O3 value might change due to mixing and photochemistry. At this stage, which is essential for the objective of this study, there is not necessarily a high correlation between PV and O3: PV might be very low but O3 still elevated due to its stratospheric origin. Importantly: this air mass is still of "stratospheric origin"! Therefore, a threshold of 1 pvu, as applied in this study, can be very misleading to identify air masses of stratospheric origin. In other words, just because PV goes below 1 pvu, the air parcel composition does not necessarily lose its stratospheric characteristics. I would find it much more meaningful to use a simulated passive stratospheric tracer to identify air of stratospheric origin and then to quantify the effects on mixing and photochemistry on ozone in these air parcels. The following papers about STT might also be helpful to the authors for further developing their methodology and for validating their results.

Lefohn, A. S., H. Wernli, D. Shadwick, S. J. Oltmans, and M. Shapiro, 2012. Quantifying the importance of stratospheric-tropospheric transport on surface ozone concentrations at high- and low-elevation monitoring sites in the United States. Atmos. Environ., 62, 646-656.

Škerlak, B., M. Sprenger, and H. Wernli, 2014. A global climatology of stratosphere-troposphere exchange using the ERA-Interim dataset from 1979 to 2011. Atmos.

Chem. Phys., 14, 913-937.

Škerlak, B., S. Pfahl, M. Sprenger, and H. Wernli, 2019. A numerical process study on the rapid transport of stratospheric air down to the surface over western North America and the Tibetan Plateau. Atmos. Chem. Phys., 19, 6535–6549.

---

## Author Comment (AC1) · 20 Aug 2019

Response to Referee Comment 1 by Anonymous Referee #1

Itahashi et al. (2019) investigated the impacts of stratospheric intrusion on tropospheric ozone based on the relationship between potential vorticity and relative humidity. They found high surface O3 are often associated with emissions whereas stratospheric intrusion contribute to O3 at elevated sites. The manuscript is in general well written. Below are a few comments need to be addressed.

**Reply:**

**We thank the reviewer for providing helpful and constructive comments. We have revised our manuscript according to the reviewer's comments and suggestions. We believe that these revisions address all points raised by the reviewer. Our point-by-point responses are provided below, and revisions are indicated in blue in the revised manuscript.**

General comments:

Tropopause height

In this work, tropopause is determined at 2.0 PVU. How is the model performance in simulating PV? How different the tropopause height calculated in this work from the traditional approach (e.g., WMO 1992).

**Reply:**

**It will be difficult to evaluate the simulated PV on upper layer using direct observations. Since the WRF simulations involve data assimilation of meteorological reanalysis fields (including upper level winds), via the nudging technique, we believe that our model-based PV should generally represent that estimated from observational data. The following figure shows the estimated tropopause altitude by PV (dynamic tropopause) and the traditional approach of WMO using the lapse rate (thermal tropopause). Both indicate a similar tropopause altitude, though some differences can be found noted in the lower latitude regions (higher altitude by PV and lower altitude by the traditional approach). To address the reviewer's question and further elaborate on the issue for other interested readers, we have included this figure in the supplemental information of the revised manuscript, and also included an additional reference to the work of Hoering et al., 1991 which investigated the tropopause altitude using two approaches. The discussion in in Section 2.1 was modified as follows in the revised manuscript:**

**"The calculation of the tropopause altitudes using PV (dynamical tropopause) and the traditional approach based on the lapse rate (thermal tropopause) defined by World Meteorological Organization (WMO) (WMO, 1992) have been reported (Hoering et al., 1991)." As shown in Figure S4, estimated tropopause altitudes averaged over April 2010 using PV in this work and the traditional approach of WMO are overall similar with below 10 km over high-latitude region and above 16 km over low-latitude region."**

[Figure]

**Figure S4: Estimated tropopause altitude averaged over April 2010 by (left) the dynamic approach using PV in this work and (right) the thermal approach using the lapse rate.**

Tropospheric O3

O3 is underestimated in the free troposphere in the model. Does the model include lightning NOx emissions? If so, are they prescribed or on-line calculated? Underestimations in lightning NOx emissions could lead to the underestimations in O3.

**Reply:**

**The reviewer raises an interesting point on the possible role of lightning NOx on the model free-troposphere O3 underestimation. In the simulations reported in this work, lightning emissions are prescribed using climatological averages as estimated by Price et al. (1997) in the GEIA database. To address the reviewer's question, we add the following clarification in Section 2.1 of the revised manuscript:**

**"The lightning emissions are prescribed using climatological averages as estimated in the Global Emission Inventory Activity (GEIA) as dataset (Price et al., 1997)."**

**As the possible reason of model underestimation for O₃, we have added the following statement in Section 3.1;**

**"In addition, the uncertainty of the lightning emissions prescribed as climatological averages in the current simulations may also contribute to the underestimation of O₃ in the free troposphere."**

Trans-pacific transport

Trans-pacific transport is not really discussed in this paper although it is shown in the title. When O3PV/O3 is used to characterize air masses, how do you distinguish air masses from trans-pacific transport?

**Reply:**

**Our sequential two papers are dedicated to the analysis of trans-Pacific transport, and we first focused on the stratospheric intrusion in this part 1 paper as high surface O₃ mixing ratio may be associated with stratospheric intrusion, in addition to trans-Pacific transport. As we have concluded in this part 1 paper, high O₃ mixing ratio is primarily related to emissions, indicating that trans-Pacific transport plays a dominant role in observed high O₃ episodes in the U.S.A. In part 2 paper, we then used sensitivity analysis technique to further perturb the emissions of O₃ precursors from East Asia and the U.S. to study the importance of trans-Pacific transport during high O₃ episodes. The two papers thus provide a comprehensive examination of the processes underlying the observed high O₃ episodes in the U.S.**

**To address the reviewer's comment, we have revised the title and the text to clarify the foci of the two parts papers as well as their relevance to "trans-pacific transport".**

Specific comments:

Figure 5, this is very complicated figure and includes a lot of information. Is there any way to evaluate PV? Regarding O3PV and O3, should they be overlapping in stratosphere that you defined based on 2PVU? Note there are some differences between these two (e.g., at Huntsville site). Any explanations on that? For observed RH profiles, in most cases, there is a steep decrease in RH from tropopause to upper layers. But at Wallops Island site, there is no such large decrease in RH, especially in early to middle April while the model shows a decreasing trend. Any explanations?

**Reply:**

As mentioned above, simulated PV could not be evaluated because of lack of direct measurement data.

As indicated in our response to comments by Reviewer #2, the $O_3$ tracer (O3PV) is used to track $O_3$ specified above 110 hPa using the O3-PV correlation and undergoes transport, scavenging, and deposition processes similar to $O_3$, but no chemical loss. This $O_3$ tracer is also initialized by the prior simulation by Hogrefe et al. (2018). Therefore, the mismatch between O3PV and $O_3$ at some sites (e.g., at Huntsville site) is related to the chemical process, higher concentration of $O_3$ rather than O3PV found near tropopause (altitude of 2 PVU) and this suggests photochemical production of $O_3$.

The steep decrease of RH is an expected general characteristic of dry stratospheric air. While this feature is apparent both in the observed and modeled RH vertical profiles at many locations, it is not clear why it is missing in the early April profiles at Wallops.

To increase the readability of this figure, we have provided more detailed discussions to help the readers to better understand the information shown in this figure.

Figure 6, the profiles (row 5) are too small. On page 11, line 5, "flight #6 might be a case of STT because observed RH is less than 10% and observed O3 mixing ratios exceed 75 ppb", where is the tropopause for this case, below or above 6km?

**Reply:**

The row 5 of Figure 6 has been expanded in the revised paper. We have analyzed the PV at this aircraft site, and the value of 2 PVU was found near 10 km. We have added the following sentences in Section 3.1.

"the tropopause as diagnosed by the PV = 2.0 PVU locates near 10 km"

Figure 9, how do you distinguish the impacts of horizontal transport and stratospheric intrusion?

**Reply:**

We have revised the air mass characterization technique. The estimation of stratospheric intrusion is based on the decision on sequential intrusion (Fig. 8). In this manner, STT caused by the horizontal transport can be considered.

Other comments

There are a few places with grammar errors.

**Reply:**

**We appreciate your careful checking. We have corrected all of them.**

Page 3, line 13-16, "On one hand…, on the other hand… .", split into two sentences.

**Reply:**

**We have split the sentence in two as suggested by the reviewer.**

Page 10, line 26, "over 500 ppbv at around 8 km The profiles…" these are two sentences.

**Reply:**

**Thank you for catching the typo. We have added a period to separate two sentences.**

Page 11, line 16, "Europe, the, model…" need correct

**Reply:**

**We have revised this point as follows; "Europe, the model…"**

Page 12, line 1, "lower RH…at lower latitudes (<40N) higher RH at higher latitude…" need correct grammar error
We have revised this sentence as follows;

**Reply:**

**The relation between modeled PV and both modeled and observed RH shows a slight dependence on latitude with higher RH at higher latitudes (> 60°N).**

Page 12, line 30-31, "…listed in Table 5 are based.", based on what? Incomplete

**Reply:**

**We have revised to remove Table 5, and this sentence was also removed.**

---

## Author Comment (AC2) · 20 Aug 2019

Response to Referee Comment 2 by Anonymous Referee #3

Review of the paper Modeling Trans-Pacific Transport using Hemispheric CMAQ during April 2010: Part 1. Model Evaluation and Air Mass Characterization for the Estimation of Stratospheric Intrusion on Tropospheric Ozone by Syuichi Itahashi and co-authors. The paper is the first in a series of two investigating CMAQ simulations of trans-pacific pollution transport for one month (April) in 2010. In contrast to what the main title promises the paper deals exclusively with ozone transport from the stratosphere and a method for a quantification of the stratospheric contribution to tropospheric ozone. The approach followed here to answer this question is quite interesting and generally well described. However, the study suffers from the extremely short period under investigation. The authors justify why they selected this period for their study. They claim that it was published earlier that trans-Pacific transport played an important role during this period (Uno et al., 2011, Lin et al., 2012a). However, they try to draw more general conclusions about the contribution of stratospheric ozone to the concentrations in the troposphere and the model performance. I believe these findings on biases and model skill are not well justified because the data set used for this type of evaluation is too small. I suggest that the authors investigate a longer simulation period in order to derive statistical parameters for the model performance and the stratospheric contribution to tropospheric O3. They can then still investigate April 2010 as a special case in more detail like it is done now.

**Reply:**

**We thank the reviewer for providing helpful and constructive comments. We have revised our manuscript according to the reviewer's comments and suggestions. We believe that these revisions address all points raised by the reviewer. Our point-by-point responses are provided below, and revisions are indicated in blue in the revised manuscript.**

**First, we revised the paper to fully describe why only a one-month simulation of April 2010 was conducted in this study. This is partly reinforced by other studies, but the selection was based on our analysis of monthly variation of MD8O3 in 2010. Second, we also carefully revised our manuscript to avoid generating the conclusions drawn from the results of this limited period.**

Specific major comments:
Page (P) 1, line (l) 21: You describe the bias in RH given by H-CMAQ, however RH should be a

quantity given by the driving meteorology model, which in this case is WRF.

**Reply:**

**We agree with the reviewer and removed this sentence in the revised paper.**

P 1, l 28-32: These statements are based on an investigation for April 2010 but the reader gets the impression that they have a more general validity. You should extend your evaluation period for this type of conclusions.

**Reply:**

**To remind the readers and not to overstate, we have added "during April 2010," in this sentence.**

P 2, l 17: Is it still true that the emissions in East Asia increase? There are more recent publications reporting the contrary.

**Reply:**

**The sentence was intended to convey that in recent years emissions across Asia are changing dramatically. To address the reviewer's comment, we replaced 'the recent acceleration' by 'the dramatic variation'.**

P 3, l 1/2: How can a publication from 1999 say something on real trends until 2010? You need to point out that this was a model study looking into the future. In addition, it would be interesting to know if the predictions for the development of Asian emissions were correct.

**Reply:**

**We revised to explicitly state that this estimation is based on the model simulation. The revised sentence in Section 1 is as follows:**

**"The global model simulation assuming the tripling of Asian anthropogenic emissions from 1985 to 2010 indicated an increase in $O_3$ mixing ratios by 2-6 ppbv in the western**

**U.S.A. and by 1-3 ppb in the eastern U.S.A. on a monthly-mean basis, with the maximum effect occurring in April-June; this increase was suggested to more than offset the benefits of 25% domestic reduction in the western U.S.A. (Jacob et al., 1999).”**

**Based on the EDGAR emission inventory, we checked this assumption and found that this was reasonable. This info was added in Section 1 as follows:**

**“Based on the Emission Database for Global Atmospheric Research (EDGAR) version 4.3.1, anthropogenic emissions of $NO_x$ and VOCs in China are estimated to have increased by 3.2 and 2.1 times during 1985-2010, respectively (Crippa et al., 2016), which is generally consistent with the assumption by Jacob et al. (1999).”**

P 3, l 8/9: When Lin et al. modelled May/June 2010 and you do April 2010: Why don't you extend your model period and put the results in perspective to their results?

**Reply:**

**The reason to focus on April 2010 is based on the analysis of monthly mean and percentiles behavior of observed MD8O3 during 2010. To address the reviewer's comment, we have added this analysis summarized in Figure S2 in the supplemental material along with our justification (see below) to support our selection of this month for model simulation.**

[Figure]

Figure S2. (Top) Monthly mean and percentiles of MD8O3 on 2010. (Bottom) Number of total observations (black color, left-axis) and exceedance of NAAQS (dark red color, right-axis; 75 ppbv is used as a criterion as 2010) on 2010.

The additional sentence in Section 1 is as follows:

"The variation in monthly mean and percentile distribution of observed MD8O3 during 2010 are shown in Fig. S2 in the supplemental material. Although high MD8O3 concentration for the 95[th] percentiles and the number of NAAQS exceedances were found during summer time, it is also apparent that mean MD8O3 during April 2010 was higher than any other month. Lower MD8O3 concentrations for the 5[th] and 25[th] percentiles were also noted as comparatively high during April 2010, indicating widespread enhancement of low-level $O_3$ further suggesting the possible impacts of trans-Pacific transport on $O_3$ levels across the U.S.A. during this month."

P 3, l 19-21: Because this is the case you need to cover other seasons with your model in order to evaluate it properly.

Reply:

This sentence conveys the general information, and we think the proper evaluation of stratospheric impacts is difficult even with a longer model simulation.

P 3, l 29/30: This objective is not covered in this paper at all. You should say more clearly what the objective of this paper is.

Reply:

To clarify the specific objectives of this Part 1 manuscript, we have revised the discussion in Section 1 as follows:

"The objective of this study is to better understand the relative contributions of precursor emissions from East Asia and the U.S.A. because the trans-Pacific transport has been recognized as an important factor. Previous studies primarily focused on Asian impacts on the western U.S.A., while this study investigates impacts across the entire U.S.A. In addition, some stratospheric intrusion events have been reported during spring 2010 (Lin

et al., 2012b), therefore this period is suitable to examine not only trans-Pacific transport but also stratospheric intrusion, both processes may contribute to the observed high $O_3$ episodes in the U.S.A. Examination of the impacts of both processes will shed light on the formation mechanisms underlying such high $O_3$ episodes, thus improving our understanding of their relative importance in leading to these high $O_3$ episodes. The results of this work will be presented in two parts. Part 1 paper focuses on characterizing the influence of stratosphere-troposphere transport on $O_3$ distribution in the lower to middle troposphere. A sequential Part 2 manuscript focuses on the contributions of emissions leading to higher $O_3$ mixing ratio through Trans-Pacific transport."

P 5, l 1 / 2: You state that O3 in the stratosphere is parameterized based on PV from WRF and an O3-PV function from Xing et al. (2016). Could you say a few words about how accurate this parameterization is?

**Reply:**

By introducing this $O_3$/PV parameterization, Xing et al. (2016) demonstrated that the parameterization improved $O_3$ model performance in the UTLS both in terms of magnitude and seasonality. The revised explanation is as follows:

"To account for the seasonal, latitudinal, and altitude dependencies in the $O_3$-PV relationship, a dynamic $O_3$/PV function was developed to consider latitude, altitude, and time based on 21-year ozonesonde records from the World Ozone and Ultraviolet Radiation Data Centre (WOUDC) and corresponding PV values from WRF-CMAQ simulations across the northern hemisphere from 1990 to 2010 and is used in H-CMAQ (Xing et al., 2016). This parameterization of $O_3$/PV is constructed at three vertical levels of 58, 76, and 95 hPa fitted as a 5th order polynomial function, and applicable in the range of 50 to 100 hPa. Based on this new parameterization, it was demonstrated that UTLS $O_3$ agreed much better with observation in terms of its magnitude and seasonality (Xing et al., 2016). Mathur et al. (2017) further demonstrated improvements in representation of seasonal variations in surface $O_3$ using the parameterization"

P 5, l 17-19: Why did you simulate only such a short period? Is it computationally expensive to run the model? Which boundary conditions of those reported in the Hogrefe et al. (2018) paper did you use?

**Reply:**

**Please refer our reply to your comment on P3, l 8/9 about the reason for the selected simulation period.**

**Regarding the computational burden, the part 2 paper uses the higher-order decoupled direct method (HDDM) to calculate the sensitivities. Although the HDDM is a sophisticated method to accurately derive sensitivities, the computational burden is much large. This is another reason to limit one-month simulation.**

**We added the description of boundary conditions as follows:**

**"The boundary conditions of H-CMAQ are taken from the clean tropospheric background values with updates to the physical and chemical sinks for organic nitrate species (Mathur et al., 2017)."**

P 8, l 13-30: Again, given these significant deviations between model results and observations, it would be beneficial for your interpretation if you extend the modelled time period.

**Reply:**

**We agree that a longer simulations period would enable more robust conclusions applicable over broader periods. However, as we mentioned above, this study focused on April 2010 when the increase of widespread MD8O3 was observed. For this one-month simulation, we prepared a suite of available data for model evaluations, including surface, vertical profile, and satellite observations.**

P 10, l 8/10: Doesn't this suggest that the scaling approach is not accurate enough for modelling ozone concentrations in the upper troposphere. So isn't there a need for adding a model component that covers stratospheric ozone with its entire chemistry and dynamics?

**Reply:**

**As we have already stated, the extension of model top layer beyond 50 hPa may be needed. We have revised this sentence to explicitly convey this meaning as follows:**

> **"Using a finer vertical resolution for the upper layers and extending the model top beyond 50 hPa to cover larger portions of the stratosphere could be potential strategies to address this need."**

P 10, l 19-21: What could be the reason for this positive bias if it occurs despite nudging of RH from reanalysis data?

> **Reply:**
>
> **Such positive bias was also found in AQMEII project despite the nudging on reanalysis data. Because RH is a diagnostic quantity which is dependent on a number of prognostic variables and sensitive parameters. Within our best knowledge, we cannot determine the critical reason to this.**

P11, l 10-20: The comparison of the ozone profiles to the model values (Table 3) suggest that the model gives too low O3 concentrations, in particular in the free troposphere. You also state this in l 9/10 on P 11. However, in Figure 7 we see a mostly positive bias with too high modeled column values, in particular over the continents, where the O3 soundings were performed. Could you explain this? Does it tell me that satellite observations deviate quite much from ozone soundings? And does it mean that your findings whether the model is too high or too low depends on the observations you compare it with?

> **Reply:**
>
> **Regarding satellite observation, Ziemke et al. (2006) reported scattered correspondence of satellite derived column ozone to ozonesonde dataset, with slight positive bias on satellite data. We have added this statement as follows:**
>
> **"In addition, the model underestimation especially in the free-troposphere is noted through comparison with ozonesonde measurements (Table 3); however, this comparison showed model overestimation. The evaluation of satellite data compared to ozonesonde exhibited scattered correspondence and slight overestimation by satellite derived column $O_3$. Therefore, the model performance could differ from that for column $O_3$."**
>
> **From Table S1 in supplemental material, the negative biases are found at > 60°N sites, but satellite data are not available over this latitude.**

P 13, l 16-18: This is another example for the main problem of this study: You investigated April 2010, only, but you give the impression that you could draw more general conclusions out of it. You should extend the modeled time series in order to give these conclusions.

**Reply:**

**To address the reviewer's comment, we have caveated our conclusions by adding "during April 2010," to this sentence.**

P 15, l 21-24: Which measure did you use for saying the model has good skill for representing the main hemispheric O3 distribution? The model is obviously too high over Africa in the equator region and it shows higher values over continents and lower values over oceans compared to the satellite observations.

**Reply:**

**This was based on the statistical analysis summarized in Table 3. To avoid the overstatement, we have revised this sentence as follows in Section 4:**

**"The results of the statistical analysis for tropospheric column $O_3$ are also listed in Table 3. The mean of observed and modeled tropospheric column $O_3$ across Northern Hemisphere is close on average, with an R of 0.65, an NMB of 4.7%, and an NME of 13.5%. The performance of tropospheric column $O_3$ judged based on the evaluation protocol developed for mixing ratios, suggests that the model satisfies the performance criteria proposed by Emery et al. (2017)."**

Minor comments:

P 2, l 16: do you mean that the number of low O3 days increased or the concentrations on the low O3 days? These are very different things and it is not clear, here.

**Reply:**

**We have revised this sentence as follows:**

**"$O_3$ concentrations on low $O_3$ days have increased"**

P 14, l 31: shows

**Reply:**

**We have corrected this.**

P 15, l 31/32 and P 16, l 8: which emissions lead to high surface O3 mixing ratios?

**Reply:**

**The part 2 paper addresses this question. To address the reviewer's comment, we have added the sentence as follows in Section 4:**

**"The Part 2 paper will focus on other factors that affect surface $O_3$ mixing ratio, namely emissions, and also examine the relative importance of $NO_x$ and VOCs."**

P 17 – 22: The references need to be revised with respect to formatting and initials.

**Reply:**

**We have rechecked reference style and revised.**

Figures:
Figure 2: It is impossible to judge the distribution of the red and the green points when the blue squares are plotted in this way.

**Reply:**

**We have revised this figure to show each surface observation as follows.**

[Figure]

Figure 4 and Figure 5: These plots look nice but I think not all of them are needed. You may put some of them into the supplemental material.

**Reply:**

**To address this comment, we have divided six sites shown in Figures 4 and 5 into 3 sites (Trinidad Head, Boulder, and Huntsville) in the main text and 3 sites (Hilo, Wallops Island, and Rhode Island) in the supplemental information.**

Figure 8: It is not clear to me what the exponential fit stands for. Is it used somewhere else?

**Reply:**

**We have revised the discussion using RH-PV, and now removed this figure.**

---

## Author Comment (AC3) · 20 Aug 2019

Response to Referee Comment 3 by Anonymous Referee #2

This manuscript is the first part of at least two parts of a paper series dedicated to the analysis of trans-Pacific transport. This first part is focused on the evaluation of the WRF / H-CMAQ model configuration and on the analysis of stratospheric intrusion. The thorough analysis in the manuscripts has two flaws:

First, the model simulation uses a horizontal grid spacing of 108 km, which is a very coarse resolution to realistically simulate stratospheric intrusions.

Second, unfortunately, the authors are omitting vital information about their most important diagnostic tool, the O3PV tracer. On the definition provided in the article the diagnostic method described in Sect. 3.2 seem to be not fully applicable and thus I doubt the results of Sect. 3.3.

Therefore, depending on the real definition (in contrast to my unterstanding of the description in the manuscript) of O3PV I am rating the manuscript as either reject or major revisions.

**Reply:**

**We thank the reviewer for providing helpful and constructive comments. We have revised our manuscript according to the reviewer's comments and suggestions. We believe that these revisions address all points raised by the reviewer. Our point-by-point responses are provided below, and revisions are indicated in blue in the revised manuscript.**

**First, we agree with the reviewer that a finer grid resolution is optimal to simulate STT but this is only possible when computational resources are available, which is unfortunately not the case for this work. The use of a grid resolution of 108 km in simulating STT used in this work is indeed consistent with the findings of other recent studies (i.e., Gray, 2003; Cristofanelli et al., 2003) suggested by the reviewer (see below our responses to major issues on this). Despite a coarse horizontal resolution used for this H-CMAQ modeling system, our model evaluation of $O_3$ concentration against surface observation network, ozonesonde, and satellite data shows an overall good model performance. Based on the good model performance and the finding from your suggested references, we think that at this time using the resolution of 108 km provides a good compromise between numerical accuracy and computational constraints. We do however acknowledge that as more computational resources become available such investigations should strive to use finer resolution in models. To address the reviewer's comment, we have added some discussion about the justification of the horizontal grid resolution used in this work along with the uncertainties associated and cited the two suggested papers.**

**Second, we have revised the paper to add all necessary information about O3PV. In particular, in the revised manuscript we further explain that the O3PV tracer tracks $O_3$ scaled to PV in the upper model layers. The O3PV tracer undergoes the same transport, scavenging, and deposition processes as $O_3$, but its mixing ratios are not affected by chemical production or loss processes Thus the O3PV can be used as a qualitative an indicator of $O_3$ of stratospheric origin as parameterized by the modeled $O_3$-PV correlation.**

**We provide below our point-by-point responses.**

Major Issues
• p. 4 l. 15-20: Looking at the very coarse horizontal resolution of 108 km, it might be nice, that the 44 layer version represents STT better than the 35 layer version. However, the horizontal resolution is much too coarse to expect a good representation of the downward mixing during STT events. (e.g., Gray 2003, Cristofanelli et al., 2003). This alone compromises the usefulness of this study.

**Reply:**

**We have carefully reviewed the two suggested references. While both indicated the models' difficulty in simulating STT at a grid resolution of >1°×1°, they show good skills when a grid resolution of 1°×1° or finer (e.g., 0.5°×0.5°) was used, which is consistent with a grid resolution of 108 km ×108 km used in this work.   Based on the findings of these two papers and the overall good performance of our model application, we believe that a horizontal grid resolution of 108 km adopted in the current H-CMAQ is adequate (though not the best) to simulate STT. To address the reviewer's comment, we revised our manuscript by providing some discussion on why this grid resolution was used and what the associated uncertainties may be in Section 2.1 as follows:**

**"While the use of finer horizaontal grid spacing can better resolve the STT processes, it will substantially increase computational demands. Cristofanelli et al. (2003) analyzed STT by combining analysis of data from a measurement network and predictions from total of seven model simulations over Europe, and reported that three models with 1°×1° horizontal resolution were able to capture the STT whereas other models with coarser resolutions were not. Another study over Europe investigated the cross-tropopause transport in terms of resolution and diffusion coefficient using horizontal resolutions of 2°×2°, 1°×1°, and 0.5°×0.5° and showed that the simulation with the 2°×2° resolution has difficulty to capture**

the tracer amount across tropopause (Gery, 2003). Based on these findings and the model evaluation results (see Section 3.1) in this work, we believe that using a grid resolution of 108 km provides a good compromise between numerical accuracy and computational constraints."

"As indicated in Mathur et al. (2017), the 44 layer configuration employed in the H-CMAQ configuration helps better capture dynamics in the vicinity of the tropopause and reduce excessive diffusion relative to coarser vertical resolution configurations."

• p. 4 l. 25: "The value of PV generally increases with altitude...": depending on the shape of the stratospheric intrusion / the PV streamer this is precisely not necessarily the case.

Reply:

Here we discuss the general feature of PV rather than the special case of such STT events. To explicitly state that, we have revised this sentence as follows in Section 2.1:

"The value of PV itself generally increases with altitude, …"

• p. 4 l. 30 - p. 5 l. 4: The definition of the O3PV tracer is not clear. How is this tracer initialised? When (at initialisation, each step ...) and where (free troposphere, stratosphere ...) is this O3-PV relationship used to define the O3PV tracer and how (is O3PV set to O3 in respective regions)? All this is essential for the information this tracer is carrying, thus a much more detailed explanation is required here.

Reply:

Similar to the chemical concentration field, the O3PV tracer was also initialized on March 1, 2010 using the prior model simulation conducted by Hogrefe et al. (2018). The O3/PV relationship is applied to estimate $O_3$ in layers above 110 hPa, specifically in the three topmost layers at 58, 76, and 95 hPa described in Xing et al. (2016). The chemically-inert O3PV tracer is set to this parameterized $O_3$ concentration in these three layers. It undergoes the same transport, scavenging, and deposition processes as $O_3$, but its mixing ratios are not affected by chemical production or loss processes and do not have any other source term beyond the parameterized values in the three topmost model layers. To address the reviewer's comment, we have revised the explanation of O3/PV relationship and O3PV tracer as follows in Section 2.1:

"To account for the seasonal, latitudinal, and altitude dependencies in the O₃/PV relationship, a dynamic O₃/PV function was developed to consider latitude, altitude, and time based on 21-year ozonesonde records from the World Ozone and Ultraviolet Radiation Data Centre (WOUDC) and corresponding PV values from WRF-CMAQ simulations across the northern hemisphere from 1990 to 2010 and is used in H-CMAQ (Xing et al., 2016). This parameterization of O₃/PV is constructed at three topmost vertical levels of 58, 76, and 95 hPa fitted as a $5^{th}$ order polynomial function, and applicable between the range of 50 and 100 hPa. Based on this new parameterization, it was demonstrated that UTLS O₃ agreed much better with observation in terms of its magnitude and seasonality (Xing et al., 2016). Mathur et al. (2017) further demonstrated improvements in representation of seasonal variations in surface O₃ using the parameterization. To track stratospheric air masses, the O₃ estimated using the O3/PV relationship in the three layers listed above is also added as a chemically-inert tracer species in the H-CMAQ simulations as O3PV tracer. The O3PV tracer undergoes the same transport, scavenging, and deposition processes as O₃, but its mixing ratios are not affected by chemical production or loss processes."

• p. 7 l. 2/3 What about high-PV structures in the free troposphere? Are they simply declared to be stratospheric?

**Reply:**

**Based on the criteria to use 2 PVU as the tropopause in this study, this case is regarded as stratosphere and not used to calculate column O₃.**

• p. 9 l. 2-4: "Generally, O3 and O3PV mixing ratios are very similar in the upper layers, especially above the 2.0 PVU line, indicating that O3 mixing ratio in these layers are dominated by stratospheric air mass." I thought that is the definition of the O3PV tracer, how could these tracers not be very similar?

**Reply:**

**We have clarified the discussion by adding the following to the discussion on Figure 5:**

**"As shown in Fig. 4, O₃ and O3PV show similar variation in the upper model layers; however, O₃ is greater than O3PV near the tropopause indicated by 2.0 PVU, and this suggests the presence of photochemical production near the tropopause."**

• p. 10 l. 9/10: more importantly the horizontal resolution needs to be increased.

**Reply:**

**As indicated in our response to earlier comments, we agree that as computational resource constraints reduce, finer horizontal resolutions should be employed in models to better capture dynamics through the troposphere and the UTLS. To address the reviewer's comment, we have revised the manuscript to discuss the uncertainties associated with the grid resolution used in this work as follows:**
**"This is the hemispheric modeling system but the finer horizontal resolution will be another way to improve this."**

• p. 10 l. 11-21: What do you expect? RH is a diagnostic quantity which is dependent on a bundle of prognostic variables and sensitive parametrisations. Thus RH is a very difficile variable to base further analysis on.

**Reply:**

**Yes, we understand that RH is a diagnostic variable. It, however, can be used as another indicator for the stratospheric air mass, as demonstrated in our discussion. To address the reviewer's comment, we added the following on Section 3.1 before the discussion on RH:**

**"Although RH is a diagnostic variable, it may also provide an indication of stratospheric air masses and is thus included in the model evaluation."**

**We reconsidered the air mass characterization technique, and RH is not used to diagnose the stratospheric air mass.**

• p. 10 l. 22-30: You show here that RH is far from realistic in the model but still the new analysis method in 3.2 is based on this diagnosed quantity?

**Reply:**

**Based on the reviewer's comments and the short comment by Heini Wernli, we have removed the use of the RH-PV relationship to estimate the stratospheric air mass in the**

**reworked analysis presented in the revised manuscript.**

• p. 11 l. 10-20 / Fig. 7: I can not agree, that the model captures the observation well. The only thing that is correct is the location of the maximum over the Pacific Ocean.

**Reply:**

**To avoid the overstatement, we have revised this paragraph as follows:**

**"The observed and modeled tropospheric column $O_3$ are compared in Fig. 7. The observed latitudinal gradients in tropospheric column $O_3$ with values greater than 40 D.U. over mid-latitudes, column values around 30 D.U. over high- and low-latitudes, and values below 20 D.U. over the Pacific Ocean near the equator are captured well by H-CMAQ. To illustrate the differences between observations and simulations, the normalized bias is also shown in Fig. 7. This normalized bias map shows model tropospheric column $O_3$ overestimation over Russia and Africa and a slight underestimation over the Pacific Ocean. While the comparison with surface observations from WDCGG shows model underestimation at four sites over eastern Europe, the model slightly overestimates tropospheric column $O_3$ in this region. In addition, the model underestimation especially over free-troposphere is found through the model evaluation with ozonesonde (Table 3); however, this comparison showed model overestimation. The evaluation of satellite data compared to ozonesonde exhibited scattered correspondence and slight overestimation by satellite derived column $O_3$. Therefore the model performance could differ from that for column $O_3$. The results of the statistical analysis for tropospheric column $O_3$ are also listed in Table 3. The mean of observed and modeled tropospheric column $O_3$ across Northern Hemisphere is close on average, with an R of 0.65, an NMB of 4.7%, and an NME of 13.5%. The performance of tropospheric column $O_3$ judged based on the evaluation protocol developed for mixing ratios, suggests that the model satisfies the performance criteria proposed by Emery et al. (2017)."**

• Sect. 3.2
– p. 11/12 / Fig. 8 / Table 5: From the data provided here, I can not agree to the method how the relationship between PV and RH is established. There is no proof, that the exponential fit is the best one. Table 5 does not provide any statistical measures to assess the quality of this fit. Maybe an elephant might have been an option too?
– I can think of low humidity conditions without stratospheric influence (e.g. above deserts).

– p. 12, ll. 9ff.: How do you deal with high-PV structures in the troposphere. Where is the tropopause diagnosted in these cases?

**Reply:**

**In the revised manuscript, we have revised the air mass characterization technique, and no longer use relationship between PV and RH is not used anymore. The revised technique classifies a stratospheric intrusion in the case of weak photochemistry (calculated by O3PV/O₃) as shown in the flowchart of Figure 8.**

– p. 12/13: too understand this method it is essential to understand how the O3PV tracer is initialised. As explained above, the description provided in this manuscript is not self-explanatory. I assume: the O3PV tracer is set every time step to O3 where PV is higher than 2 PVU (this might include high-PV structures in in the troposphere) and might blur the signal of "real" stratospheric air.:

**Reply:**

**At initialization, 3-dimensional fields of the O3PV were derived from prior H-CMAQ simulations described in Hogrefe et al. (2018). Since the O3PV tracer was included to qualitatively track O₃ from the model upper layers (nominally representative stratospheric origin) as parameterized by the PV-O₃ correlation, the tracer's mixing ratio in layers only above 110hPa is set every time step based on the PV-O₃ scaling. More details on the definition of the O3PV tracer are provided in our response to the comment on "p. 4 l. 30 - p. 5 l. 4"**

– Additionally, as the O3PV tracer is transported and depositioned due to its own gradients many deviations between the Ozone and the O3PV tracer might be caused by differences in transport and sinks and not in photochemistry.

**Reply:**

**As explained in the manuscript, the O3PV tracer undergoes the same transport (advection, turbulent mixing, and clouds), scavenging and deposition processes as O₃. The only difference is that it does not undergo any chemical transformation, and thus the differences between modeled O₃ and O3PV fields can be attributed to the photochemistry which impacts the simulated O₃ but not O3PV.**

– Fig. 10 and corresponding text: The description of you results reads as if stratospheric ozone would be inert and only tropospheric ozone would take place in photochemistry. The fastest process of all are the autocatalytic cycles of ozone production and destruction. Therefore, the amount of stratospheric ozone influences directly the photochemisty. How is this stratospheric ozone mass calculated? Is it the integral over O3PV? In that case, I would say that the assessment is wrong as you miss its photochemical sink. (provide more details about the calculation p.13, ll.12-14)

– From the current knowledge about the method I would say, that a continuously initialised stratospheric tracer could be a diagnostic tool to diagnose stratospheric influence. But the quantification diagnostic introduced in Sect. 3.2. does not work, unless the authors omitted to provide a lot of vital information about their method.

• Sect. 3.3: As I question the diagnostic method explained in Sect. 3.2, I have to doubt the results of this section as well. Of course you can say, whether the air is influenced by stratospheric air, but the percentages provided in Fig. 11 mean nothing.

• p. 15 l. 30-33: Due to the coarse horizontal resolution of the model it was not to be expected that stratospheric ozone is transported downward efficiently enough to reach the surface.

**Reply:**

**Chemistry is still active across the modeled vertical extent. As in other studies utilizing scaling of $O_3$ based on PV values, here too $O_3$ in the model's UTLS is scaled to the dynamically evolving PV fields. The O3PV is an additional diagnostic tracer added to qualitatively track the influence of this $O_3$ originating in the model upper layers (representing stratospheric $O_3$ in the absence of a complete representation of the stratosphere and it's chemistry) through the modeled vertical extent, with special interest on the amounts in the boundary layer. We hope that this along with the changes related to the tracer description described in response to earlier comments helps better explain the configuration, processed modeled, and the interpretation of the modeled fields of O3 and O3PV.**

**To address these four comments and questions, and also to take into account the short comment raised by Heini Wernli, we have revised the air mass characterization technique for stratospheric air mass characterization.**

**The flowchart depicted in Fig. 8 is revised to exclude the judgement based on PV-RH relation according to the comment by Heini Wernli. The way to judge stratospheric air mass is limited to O3PV/O3 near 1.0 (range between 0.9 and 1.1), which indicates the weak photochemistry and possible impacts by stratospheric air mass. Then, the top layer (z=44)**

**is defined as stratospheric air mass, and if the above grid cell is judged as stratospheric air mass under the weak photochemistry, the grid is determined as stratospheric air mass. Figs. 9, 10, and 11 have been updated according to this revised method to estimate the stratospheric air mass. Fig. 8 and Table 5 have been removed because now we do not need PV-RH relationship to judge the stratospheric air mass.**

Minor Issues
• title: should contain the model version, as evaluations are always specific for the used model version. Additionally, the title is misleading as the authors miss to point out the interdependencies between STT and trans-Pacific transport.

**Reply:**

**We think the model version in the title may not be needed, but we have revised the title to explicitly indicate both trans-Pacific transport and STT.**
**We however, provide model versions and details on specific addition in the manuscript description.**

• p. 1 l. 29/30: not clear what the message is. Where else could STT impacts come from?

**Reply:**

**To address the reviewer's concerns we have clarified the discussion in the abstract as follows:**
**"Over the U.S.A., STT impacts show large day-to-day variations, and STT impacts can either originate from the same air mass over the entire U.S.A. with an eastward movement found during early April, or stem from different air masses at different locations indicated during late April."**

• p. 2 l. 1: as STT is event based I doubt that the impact is near constant.

**Reply:**

**We have revised previous Figure 12 (now Figure 10) to use monthly-mean data, and explicitly mention that the analysis is based on monthly means. Then we showed Figures 11 and 12 to show the daily variation of stratospheric intrusion over the U.S.A.**

• p. 2 l. 17: "acceleration of anthropogenic emissions" ? emissions are not accelerated. They might increase and their increase might be accelerated ...

**Reply:**

**We have revised this sentence as follows in Section 1:**
**"the dramatic variation of anthropogenic emissions in East Asia"**

• p. 4 l. 21: What is cb05e51? A GIT tag ?

**Reply:**

**This statement stands for EPA modifications implemented in CMAQ version 5.1. This is the actual chemical mechanisms available in CMAQ version 5.1 and later, and is documented in the release notes and model documentation web page (https://www.airqualitymodeling.org/index.php/Cb05e51_species_table). We would like to keep this expression.**

• unify "O3/PV" vs. "O3-PV" relationship.

**Reply:**

**We have unified this wording in "O3/PV" throughout the manuscript.**

• Sect. 2.1: Are these (WRF and H-CMAQ) continuous simulations or are they re-initialised?

**Reply:**

**WRF is newly simulated from March 2009 to have more than one-year spin-up time. H-CMAQ is initialized on March 1 based on concentration and tracer fields archived from a longer H-CMAQ simulation described in Hogrefe et al. (2018). In the current study, the H-CMAQ simulations start on March 1; we use the entire month of March as additional spin-up and then, April is used for the analysis period.**

• Sect. 2.2.3 It is really necessary to talk about un-used flight data?

**Reply:**

**This information was provided for completeness as we only used such data at one site. Readers may be curious why we did not use the data at the other two sites.**

• Fig. 1: Usage of lighter colors would make it easier to see the symbols. The grey aircraft symbol is hard to distinguish from the grey map lines.

**Reply:**

**The gray color has changed into light-blue color in Figure 1.**

• longitude / latitude information is missing in all maps

**Reply:**

**We have prepared the map indicating longitude and latitude of H-CMAQ modeling domain in Figure S3 in the supplemental material.**
**We believe including longitude and latitude would lead to busy figures; therefore, we would like to omit longitude/latitude information in Figures 1, 3, 7, and 11.**

• Fig. 4: thick line not identifyable, black lines are distinguishable only at 300 % zoom and more.

**Reply:**

**To be consistent with Figure 5, we have changed the color of thick lines (2.0 PVU) from black to red, and we also enhanced the thickness of other black lines.**

• Table 1: The tables content is not understandable without providing more details, e.g.:
– What does "ranged" and "zero-out" mean?

**Reply:**

**To be consistent with another expression, we have revised "ranged 10-25 ppbv" into "10-**

**25 ppbv".**

**We also have added the footnote to explain "zero-out" as follows:**

**"ᵃ: Estimate the impact from the difference between the standard simulation and a simulation with eastern Asian anthropogenic sources shut off."**

– "tagged O3": which tagging method?

**Reply:**

**We have added the footnote for detail information of "tagged $O_3$" as follows:**
**"ᵇ: This tagged method divides simulated $O_3$ into individual $O_3$ tracer to track $O_3$ produced in different region."**

– "tropopause tracer": How defined / initialised?

**Reply:**

**We have added the footnote for detail information of "tropopause tracer" as follows:**
**"ᶜ: This tracer method accounts for STT contribution to $O_3$ using e90 tracer, which differentiates tropospheric air mass based on the globally uniform surface source and 90-day folding lifetime; both have been spun-up for three years."**

– Table 1: the descriptions of the "Estimated impacts" are completely messed, e.g., "5-7 ppbv (17 April- 15 May 2006; INTEX-B), increased by 1-2 ppbv from April-May 2000" What does this mean? The estimate stems from a measurement in 2006 during the INTEX-B compaign and is compared to a 2000 value, where we do not know anything about? And do you mean that it impact increased by 5-7ppbv?

**Reply:**

**The original table includes two types of information derived by different methods, hence we have revised this row into two rows to clearly indicate the method description.**

Literature:
• Cristofanelli, P., Bonasoni, P., Collins, W., Feichter, J., Forster, C., James, P., Kentarchos, A., Kubik, P., Land, C., Meloen, J., Roelofs, G., Siegmund, P., Sprenger, M., Schnabel, C., Stohl, A., Tobler, L.,

Tositti, L., Trickl, T., and Zanis, P.: Stratosphere-to-troposphere transport: A model and method evaluation, J. Geophys. Res., 108, 8525, doi:10.1029/2002JD002600, 2003.

• Gray, S.: A case study of stratosphere to troposphere transport: The role of convective transport and the sensitivity to model resolution, J. Geophys. Res., 108, 4590, doi:10.1029/2002JD003317, 2003.

---

## Author Comment (AC4) · 20 Aug 2019

**Response to Short Comment by Heini Wernli**

I share the concerns raised by the 2nd reviewer about model resolution and the description of the setup of the stratospheric tracer O3PV. In addition, I would like to comment on the issue of using PV to identify "air of stratospheric origin". According to Fig. 10, an air mass is classified as a strong/moderate/weak stratospheric intrusion if PV is larger than 3/2/1 pvu. Here it is not fully clear what "strong" and "moderate" mean: if air with PV>2 pvu is stratospheric then why calling PV>3 pvu "strong intrusion"? It has higher PV but this paper is about ozone. This measure of "strength" does not necessarily reflect the influence on ozone. Things become more problematic in the text where it reads: "At least, a PV value of 1.0 PVU and corresponding RH (from Table 5) are required to judge stratospheric origin." Note that "stratospheric origin" is not the same as a "stratospheric intrusion". An originally stratospheric air parcel that experiences STT, first has high PV (> 2pvu) and, most likely, high O3. It then loses PV due to some diabatic process (turbulence, radiation, convection) and enters the troposphere (what we call STT). Ozone values might still be high. Entering further into the lower troposphere many things can happen: diabatic processes can further reduce PV (to very low values of less than 1 pvu), the air parcel may become moister (due to turbulent mixing) and its O3 value might change due to mixing and photochemistry. At this stage, which is essential for the objective of this study, there is not necessarily a high correlation between PV and O3: PV might be very low but O3 still elevated due to its stratospheric origin. Importantly: this air mass is still of "stratospheric origin"! Therefore, a threshold of 1 pvu, as applied in this study, can be very misleading to identify air masses of stratospheric origin. In other words, just because PV goes below 1 pvu, the air parcel composition does not necessarily lose its stratospheric characteristics. I would find it much more meaningful to use a simulated passive stratospheric tracer to identify air of stratospheric origin and then to quantify the effects on mixing and photochemistry on ozone in these air parcels. The following papers about STT might also be helpful to the authors for further developing their methodology and for validating their results.

Lefohn, A. S., H. Wernli, D. Shadwick, S. J. Oltmans, and M. Shapiro, 2012. Quantifying the importance of stratospheric-tropospheric transport on surface ozone concentrations at high- and low-elevation monitoring sites in the United States. Atmos. Environ., 62, 646-656.

Škerlak, B., M. Sprenger, and H. Wernli, 2014. A global climatology of stratosphere troposphere exchange using the ERA-Interim dataset from 1979 to 2011. Atmos. Chem. Phys., 14, 913-937.

Škerlak, B., S. Pfahl, M. Sprenger, and H. Wernli, 2019. A numerical process study on the rapid transport of stratospheric air down to the surface over western North America and the Tibetan Plateau. Atmos. Chem. Phys., 19, 6535–6549.

**Reply:**

**We thank Dr. Heini Wernli for providing helpful and constructive comments.**

**In the revised manuscript, we have revised the air mass characterization technique based on your comment. The relationship between PV and RH is not used. The revised flowchart is shown in Fig. 8. The stratospheric air mass is assigned in the top layer and the stratospheric intrusion is determined in the case that the photochemistry (calculated by $O3PV/O_3$) is weak.**

**We believe that these revisions address all points raised by the reviewer, and revisions are indicated in blue in the revised manuscript.**

---

## Author Response (AR2)

**Response to Comment by Editor**

**Co-Editor Decision: Reconsider after major revisions** (27 Sep 2019) by Pedro Jimenez-Guerrero

Comments to the Author:

Dear authors:

Please address the comments by the reviewers before the manuscript can be considered for final publication.

Non-public comments to the Author:

Dear authors:

One of the reviewers raise very important concerns regarding the scientific knowledge gain and the skill of the simulations. I share some of these concerns. The limitations pointed out by the reviewer are well established and supported, so I would beg you to address them before the manuscript can be considered for final publication.

**Reply:**

**Thank you for your consideration of our manuscript and for your comments and suggestions. We have carefully considered yours and the additional comments from the two reviewers, responded to their comments and suggestions in the attached point-by-point responses, and incorporated revisions in the manuscript to address the shortcomings they noted. In particular to address the Reviewer #2's concern on the possible impacts of the model grid resolution on resolving STT dynamics and events, we conducted two additional sets of WRF model simulations over the contiguous U.S. using 36-km and 12-km grid spacing, respectively. We analyzed and compared the vertical and temporal variations in the estimated potential vorticity (PV) fields at the locations of the ozonesonde sites which are the focus of the analyses; the results are summarized in three additional figures: S7-S9 in the supplemental information and associated brief discussion in the manuscript main text. As detailed in our response to reviewer #2, we find that for simulations at the 3 grid resolutions examined (108-km, 36-km, and 12-km), time-height variations in PV at these locations are largely similar and so is the estimated altitude of the tropopause (i.e., altitude of 2PVU) which is used in our analysis. As shown by the similarity of the altitude of the 2PVU estimated across the 3 different resolutions, the interpretation of STT events in our analysis is not strongly influenced by the lack of resolution in our original 108-km calculations, as speculated by Reviewer #2. This is not completely surprising since all model calculations employed assimilation of analyzed meteorological fields (constrained by prior observations) in the model's UTLS. Consequently, the STT dynamics and capturing of specific events using the PV criteria are similar. Finally, a comparison of $O_3$ at the model top layer also show good correspondence in the magnitude of $O_3$ at the model top using**

PV estimates from the 3 different resolutions. As shown by the additional information in the new Figures S7-S9, the robustness of our results and interpretations are not strongly influenced by perceived lack in horizontal grid resolution. We thank the reviewer for raising this important consideration, addressing which has helped strengthen the analysis presented and the associated robustness of the approach and results. We believe that these additions and revisions to the manuscript now address all the points raised by the reviewers as well as your concerns.

We trust that our response to the reviewer comments and the revisions incorporated in the manuscript meet with your criteria for acceptance. Thank you for your consideration of our manuscript for publication in Atmospheric Chemistry and Physics.

Response to Referee Comment 1 by Anonymous Referee #1

The manuscript has been improved significantly after revision especially for the discussions on tropopause, O3PV tracer, and O3/PV ratios. The authors have addressed all my comments and I'm satisfied with the authors' responses. However, there are still a couple of technical issues needed to be corrected.

**Reply:**
**We thank the reviewer for providing helpful and constructive comments. We have revised our manuscript according to the reviewer's technical comments and suggestions. We believe that these revisions address all points raised by the reviewer. Our point-by-point responses are provided below, and revisions are indicated in blue in the revised manuscript.**

Technical errors:
Page 5, line 22, "..have been reported (Hoering et al., 1991)"", no need "
    **Reply:**
    **We thank the reviewer for catching the typo. The extra quote mark has been removed.**

Page 13, line 25, "These stratospheric intrusion are..."
    **Reply:**
    **We have corrected as "These stratospheric intrusions are".**

Page 14, line 28-32, "tropospheric o3 column mass" is not a very accurate term here. same for Figure 10.
    **Reply:**
    **We have revised this term as "tropospheric $O_3$ column".**

Response to Referee Comment 2 by Anonymous Referee #2

The manuscript improved a lot, thanks for that!

Nevertheless, the issue with the horizontal resolution is still a major weakness of this article. As detailed below (item 2+3), both articles cited in my first review do not, at least from my point of view, support the hypothesis of this article (top page 5), that the CMAQ model with 108 km grid spacing is able to capture SST events sufficiently. Additionally, I am not convinced that the provided evaluation shows that the SST events are captured correctly: the averaged SST maybe, but not the events. As the result can change considerably simply due to employingby decreasing the horizontal resolution, the scientific gain from this study is very, very limited. I understand, that it is an issue of computer resources, nevertheless, some studies can simply not be performed, if the resources are not available.

**Reply:**
**We thank the reviewer for providing helpful and constructive comments. We have revised our manuscript according to the reviewer's technical comments and suggestions. In particular, to address the reviewer's concern on lack of adequate resolution in our model calculations, we have now also included results from new WRF simulations over the continental U.S. employing a finer resolution to examine the possible impacts of resolution on representing dynamics for the SST events analyzed at the locations of the ozonesonde launches – the results and changes incorporated in the manuscript are further detailed in response to the reviewer's specific comments below. We believe that these revisions address all points raised by the reviewer. Our point-by-point responses are provided below, and revisions are indicated in blue in the revised manuscript.**

There are some remaining issues I'd like to point to:
1. Title: the title sounds a little bit weird. Leaving out the first words it reads "stratospheric intrusion of tropospheric ozone [...]". This seems wrong. I suggested to delete the word "tropospheric" from the title, as "trans-pacific transport" includes, at least for me, that it is a tropospheric process.

> **Reply:**
> **Following the reviewer's suggestion we have revised the title as follows:**
> **"Modeling Stratospheric Intrusion and Trans-Pacific Transport on Tropospheric Ozone using Hemispheric CMAQ during April 2010: Part 1. Model Evaluation and Air Mass Characterization for Stratosphere-Troposphere Transport"**

2. p. 4 ll. 27-30: In my opinion the results of Cristofanelli et al. are summarized in a wrong way. The important outcome is, that the models with lagrangian transport scheme were able to capture SST, but the Eulerians were not. Unfortunately, the models with lagrangian transport schemes had a resolution of 1x1 degree and the Eulerian models had a much coarser resolution. But the Cristofanelli publication does not provide any evidence, that an Eulerian model in 1x1 degree would be able to capture SST.

**Reply:**

**We thank the reviewer for this comment and have re-examined the results and discussion in Cristofanelli et al. We have revised this discussion to reflect that their results suggested that models with a lagrangian transport scheme were able to capture SST. The revised sentence is as follows:**

**"STT was analyzed by combining analysis of data from a measurement network and predictions from total of seven model simulations over Europe, and reported the advantages of lagrangian models in capturing the STT (Cristofanelli et al., 2003)."**

3. p. 4 ll. 30-32: The study of Gery (2003), as pointed out explicitly within the article, is only valid for the Unified model of the UK Met Office (UM). Gery (2003) explicitly points out that the results very much depend on the convection and diffusion parametrisation used in the respective model. Therefore one can not simply assume that all models are able to capture SST events in a horizontal resolution, just because the UM is able to do so. In principle each model needs to be evaluated from which horizontal resolution on it is able to capture SST resonable. I do not know such an evaluation for CMAQ driven by WRF.

**Reply:**

**While the results of Gery (2003) are indicative of the resolution impacts, we acknowledge that they should be viewed as results from a single model for a specific region and that the combined impacts of transport formulation, parameterizations, and grid resolution should be examined in each application.**

**To address the reviewer's concerns, we have conducted two additional WRF simulations over contiguous U.S. (CONUS) domain with 36 and 12 km grid resolutions but with all other physics configuration identical to that of the original 108 km simulation. The analyzed time-height curtain plot of PV is shown below and also included as Figure S7 in the revised supplemental material. As can be seen in these curtain plots, generally modeled PV fields at different resolutions are largely similar . To further quantify the impacts of differences across these resolutions on interpretation of SST, differences in the time-height PV profiles calculated by 36 km−108 km and 12 km−108 km are shown in Figs. S8 in the supplemental material. These results clarified that the higher (lower) PV at upper (lower)**

altitudes is enhanced (weakened) by increasing horizontal grid resolution. The difference from the 108 km simulation was much clearer in the 12 km simulation. As also shown by the similarity of the altitude of the 2PVU estimated across the 3 different resolutions, the interpretation of STT events in our analysis is not strongly influenced by the lack of resolution in our original 108-km calculations. This is not completely surprising since all model calculations employ assimilation of analyzed meteorological fields in the model's UTLS. Consequently, the STT dynamics and capturing of specific events using the PV criteria are similar. Finally, since in our chemical transport calculations we scale $O_3$ in the model's UTLS using the estimated PV fields, in Figure S9 of the revised manuscript we also present comparisons of the estimated $O_3$ in the uppermost layer based on the $O_3$/PV relation used in this study. These comparisons also show good correspondence in the magnitude of $O_3$ at the model top using PV estimates from the 3 different resolutions.

Collectively, we believe that these additional analyses should address the reviewer's concerns on the robustness of our results due to the horizontal grid resolution employed. As shown by the additional information in the new Figures S7-S9, the robustness of our results and interpretations are not strongly influenced by perceived lack in resolution as speculated by the reviewer. We nevertheless thank the reviewer for raising this important consideration, addressing which has helped strengthen the analysis presented and the associated robustness of the approach and results.

The added figures are Figs. S7- S9 in the supplemental material, and the relevant sentences in the main manuscript have been modified as follows:

"To investigate the effect of horizontal grid resolution on the representation of STT, additional WRF simulations were conducted over CONUS domain with 36 and 12 km horizontal grid resolutions, and temporal and vertical variations in simulated PV fields across the different resolutions (108, 36, and 12 km) were compared. These results are shown in Fig. S7 in the supplemental material. Generally, the modeled PV fields estimated with different horizontal grid resolutions showed similar features. The differences are displayed in Fig. S8 in the supplemental material. It was revealed that higher (lower) PV at upper (lower) altitude is enhanced (weakened) by increasing horizontal grid resolution. As expected, larger differences are noted between the 108 km and 12 km fields than those between the 108 km and 36 km fields. Although the enhancement of PV at upper altitudes could lead to increase in estimated $O_3$ through the $O_3$/PV relationship used in the model, no systematic differences are noted in the estimated $O_3$ in the model's UTLS across the three resolutions, at least at the ozonesonde observation sites where our analysis is focused. A comparison of the altitude of 2 PVU which is used to diagnose the tropopause is also plotted in Fig. S8. As noted by the similarity of the altitude of the 2PVU across the 3

different resolutions, the interpretation of STT events is not strongly influenced by the horizontal resolution employed in this study. This is because all model calculations employ assimilation of analyzed meteorological fields in the model's UTLS, resulting in comparable representation of STT events. Finally, a comparison of estimated $O_3$ at the model top-layer based on the $O_3$/PV relation (Xing et al., 2016) by using different PV simulated from different horizontal grid resolutions is illustrated through scatter-plots in Fig. S9 in the supplemental material. These comparisons indicate good correspondence in the magnitude of $O_3$ at the model top using PV estimates from the 3 different resolutions. At the Boulder site (Fig. S9 (b)), the use of finer grid resolutions could sometimes lead to higher $O_3$ concentrations. Collectively, the comparisons in Figure S7-S9 suggest that the 108 km horizontal grid resolution in H-CMAQ modeling system in conjunction with the physics and data assimilation options employed in the driving WRF model can capture the variability in the PV fields and associated STT $O_3$ impacts."

[Figure]

**Figure S7.** Curtain plots of modeled PV by (left) WRF over northern hemisphere with a 108 km horizontal grid resolution, (center) WRF over CONUS domain with a 36 km horizontal grid resolution, and (right) WRF over CONUS domain with a 12 km horizontal grid resolution at U.S. ozonesonde sites of (a) Trinidad Head (CA), (b) Boulder (CO), (c) Huntsville (AL), (d) Wallops Island (VA), and (e) Rhode Island (RI) during April 2010.

[Figure]

**Figure S8.** Curtain plots of (right) the difference of modeled PV calculated from 36 km−108 km, (center) from 12 km−108 km, and (right) modeled PV lines of 2 PVU by 108 km (red), 36 km (dark orange), and 12 km (light orange) at U.S. ozonesonde sites of (a) Trinidad Head (CA), (b) Boulder (CO), (c) Huntsville (AL), (d) Wallops Island (VA), and (e) Rhode Island (RI) during April 2010.

[Figure]

**Figure S9. Correspondence of estimated O₃ concentration based on O₃/PV relation at the uppermost layer as (left) 108 km vs. 36 km and (right) 108 km vs. 12 km at U.S. ozonesonde sites of (a) Trinidad Head (CA), (b) Boulder (CO), (c) Huntsville (AL), (d) Wallops Island (VA), and (e) Rhode Island (RI). Plots are hourly data during April 2010 (total number is 720).**

4. p. 9 ll. 19-30: Instead of mentioning the biomass burning as the first and thus most important reason, I would guess, that the underrepesentation of the STT events due to the coarse resolution is the major reason.

**Reply:**

**We have added the following sentence to indicate the horizontal grid resolution as a possible reason:**

**"Another possible reason may stem from the use of a coarse horizontal resolution."**

**Because we cannot judge the priority of these reasons (based on the analysis presented in response to comment #3), we just mention the possible reasons here.**

5. p. 11 ll. 11-13: As above, I opt that a higher resolution in the horizontal would be much more promising than increasing vertical layering. (By the way, Gery (2003) points out, that increasing vertical resolution in his simulations decreased the vertical transport, which is contrary to the statement in the present article.)

**Reply:**

**As detailed in our response to comment #3, the analysis of WRF simulations at 36-km and 12-km horizontal resolutions do not necessarily suggest significant increase in skill in the diagnosis of STT events relative to our base calculation. Based on the discussion of these results in the revised manuscript we now have removed this earlier somewhat speculative sentence: "This is a hemispheric modeling system but the finer horizontal resolution will be another way to improve this.".**

6. Section 3.2: SST events are connected with PV streamers and the stratospheric air mass is transported downward in the wake of the front, thus there is a non-negligible horizontal motion involved. I would expect that a high amount of stratospheric influenced air is discarded, just be characterizing the air column-wise from top to bottom.

**Reply:**

**As we have illustrated in our response to comment#3., the simulated dynamics associated with STT over the CONUS appear to be largely similar across the 3 different horizontal resolutions, primarily because of the assimilation of meteorological data in the UTLS. The slight enhancement of PV at higher altitude could lead to the increased $O_3$ concentrations during some times and might influence mid-tropospheric $O_3$ in some cases, though systematic differences in air mass motion were not detected based on the PV analysis.**

7. Fig.8: This flow-chart is not really a flow-chart. Somehow the information that the flowchart "cycles" over all columns or for one column only (and that the analysis is columnwise) needs to be

included. More importantly it should be indicated that, if the above grid box is denoting a stratospheric air mass, the vertical index is decrease (one box down in the column) and the analysis is continued for the next lower box. Currently, the same analysis would be repeated over and over again for the same grid box, as no index change is depicted in the flowchart. Last but not least, an outcome for each grid box, indicating stratospheric or not, should be explicitly shown.

**Reply:**

**This flowchart has been revised to explicitly show that repetition is conducted over layers from number 43 to number 1 (surface) by including the loop chart.**

Minor Issues

• p. 4 last sentence: what is the "tracer amount across tropopause"? Are you omitting the word transport here somehow?

**Reply:**

**This sentence has been revised as "the tracer transport across the tropopause".**

• p. 5, ll. 20-22: This is not a sentence.

**Reply:**

**This sentence has been revised as "The tropopause altitude can be also diagnosed by the traditional approach based on the lapse rate (i.e., thermal tropopause) defined by World Meteorological Organization (WMO) (WMO, 1992), and comparisons with that diagnosed using PV (i.e., dynamical tropopause) have been reported (Hoering et al., 1991)."**

• p. 13, ll. 3/4: "The flowchart the air mass ..." ??? Is not a sentence.

**Reply:**

[revised manuscript text omitted]

The copyright of individual parts of the supplement might differ from the CC BY 4.0 License.

[Figure]

**Figure S1. Long-term trends from 2005 to 2015 as before and after 5 years comparison to 2010. (Top) Mean (blue color, left-axis) and maximum and minimum (black color, right-axis) MD8O3 on April. (Center) Number of total observations (black color, left-axis) and exceedance of NAAQS (dark red color, right-axis; 75 ppbv is used as a criterion as 2010) on April. (Bottom) Annual NOₓ (purple) and VOCs (green) emissions in the U.S.A. except wildfire (https://www.epa.gov/air-emissions-inventories/air-pollutant-emissions-trends-data).**

[Figure]

**Figure S2. (Top) Monthly mean and percentiles of MD8O3 on 2010. (Bottom) Number of total observations (black color, left-axis) and exceedance of NAAQS (dark red color, right-axis; 75 ppbv is used as a criterion as 2010) on 2010.**

[Figure]

**Figure S3. The information of longitude and latitude in H-CMAQ modelling system.**

[Figure]

Dynamic tropopause          Thermal tropopause

Tropopause altitude [km]

**Figure S4. Estimated tropopause altitude averaged over April 2010 by (left) the dynamic approach using PV in this work and (right) the thermal approach using the lapse rate.**

[Figure]

**Figure S5.** Curtain plots of modeled (left) O₃, (center) O3PV, and (right) RH at U.S. ozonesonde sites of (a) Hilo (HI), (b) Wallops Island (VA), and (c) Rhode Island (RI). during April 2010. Yellow stars indicate the time of available ozonesonde measurements. Contour lines of modeled PV are also inserted for contours of 1.0, 1.5, 2.0, 2.5, and 3.0 PVU with thick red lines denoting the 2.0 PVU contour as an index to diagnose the tropopause. See also Figure 4.

[Figure]

**Figure S6. Vertical profiles of observed and modeled O₃ and RH at U.S. ozonesonde sites of (a) Hilo (HI), (b) Wallops Island (VA), and (c) Rhode Island (RI). Also see Figure S5 for ozonesonde measurement times. For modeled O₃ and RH, the hourly result corresponding to the ozonesonde measurement time is shown by circles, and the maximum and minimum model results within ±2 hours of the measurement time are shown by whiskers. For observed O₃ at Hilo and Boulder, the range of uncertainties of the O₃ observations is shown by whiskers. Modeled O3PV and PV are also shown. Modeled PV profiles are plotted in red, and vertical lines corresponding to a PV value of 2 PVU are inserted as an index of the tropopause, and the layer range diagnosed as stratospheric air mass is colored in purple. See also Figure 5.**

[Figure]

**Figure S7. Curtain plots of modeled PV by (left) WRF over northern hemisphere with a 108 km horizontal grid resolution, (center) WRF over CONUS domain with a 36 km horizontal grid resolution, and (right) WRF over CONUS domain with a 12 km horizontal grid resolution at U.S. ozonesonde sites of (a) Trinidad Head (CA), (b) Boulder (CO), (c) Huntsville (AL), (d) Wallops Island (VA), and (e) Rhode Island (RI) during April 2010.**

[Figure]

**Figure S8.** Curtain plots of (right) the difference of modeled PV calculated from 36 km−108 km, (center) from 12 km−108 km, and (right) modeled PV lines of 2 PVU by 108 km (red), 36 km (blue), and 12 km (sky blue) at U.S. ozonesonde sites of (a) Trinidad Head (CA), (b) Boulder (CO), (c) Huntsville (AL), (d) Wallops Island (VA), and (e) Rhode Island (RI) during April 2010.

(a) Trinidad Head

(b) Boulder

(c) Huntsville

(d) Wallops Island

[Figure]

[Figure]

(e) Rhode Island

[Figure]

[Figure]

**Figure S9. Correspondence of estimated O$_3$ concentration based on O$_3$/PV relation at the uppermost layer as (left) 108 km vs. 36 km and (right) 108 km vs. 12 km at U.S. ozonesonde sites of (a) Trinidad Head (CA), (b) Boulder (CO), (c) Huntsville (AL), (d) Wallops Island (VA), and (e) Rhode Island (RI). Plots are hourly data during April 2010 (total number is 720).**

[Figure]

**Figure S10.** Curtin plot of model-diagnosed air mass characterization for (left) O3PV/O₃ and (right) stratospheric air mass at U.S. ozonesonde sites of (a) Hilo (HI), (b) Wallops Island (VA), and (c) Rhode Island (RI) during April 2010. See also Figure 9.

[revised manuscript text omitted]

---

## Author Response (AR3)

**Editor and Reviewer Comment**: *Dear authors. I have received the following comments from one of the reviewers: "I am still not completely convinced that the amount of ozone transported down to the surface is resolution independent. Even if the PV structures in all resolutions are similar, somewhen/somewhere during the transport from the stratosphere to the lowermost troposphere the relation between PV and ozone is breaking down (see also public comment by Heini Wernli). Thus I still think the authors did not prove that the ozone transport is not resolution dependent.". Please confirm if you can correctly address this point before final consideration for publication*

**Response**: We appreciate the time and effort devoted by the reviewers in re-reviewing the revised manuscript and for considering our response to their comments. We have carefully considered the reviewer's comment and have also re-read the manuscript to ensure that we have not mis-conveyed the possible effects of model grid resolution on the interpretation of our results. As we indicated in response to comments on previous versions of the manuscript, the choice of a 108km horizontal resolution for the hemispheric version of CMAQ was based on seeking a balance with available computational resources while capturing the large-scale effects of intercontinental pollution transport and likely effects of stratosphere-troposphere exchange processes on the tropospheric $O_3$ burden. As noted, both previously in Mathur et al. (2017) and discussions in the current manuscript, as computational resources increase, improvements in both horizontal and vertical resolution should be explored to assess further improvement in representation of transport processes and model performance. We agree with the reviewer that finer grid resolution could improve representation of the 3-D transport of ozone (we already note that both in the discussion of our results and re-emphasize in the conclusions) and that "*somewhen/somewhere*" the results of model configurations with differing resolutions will be different, as would also from perturbations to model data, parameters, and model parameterizations and numerical schemes. We however respectfully disagree with the suggestion that the current resolution renders the analyses presented invalid (previous Referee#2 comment: "*some studies can simply not be performed, if the resources are not available*"), especially since no specific resolution is recommended in the reviewer comments or in previous studies examining such effects and also given the analysis of the additional WRF simulations at finer resolutions which illustrated that our method and inferences were not adversely impacted by the choice of the 108km resolution. We do however, believe that additional clarifications on the methodology and the inferences from the finer resolution WRF simulations could help provide better context for the results presented and possible next steps, and have attempted to do so in the revised manuscript.

We also believe the reviewer has likely misinterpreted the use of the PV-O3 correlation in our model calculations. As briefly summarized in Section 2.1 and detailed in Xing et al. (2016), the model $O_3$ only in grid cells above 100hPa is scaled to time and space varying PV fields using the Xing et al. (2016) parameterization. The model dynamics (3D advection, convective and resolved cloud mixing, and turbulent transport) then transport this $O_3$, nominally representative of stratospheric origin, through the rest of the model vertical extent to the surface. Please note that the PV-O3 scaling is not used to represent the "*transport from the stratosphere to the*

*lowermost troposphere*" as speculated in the reviewer's comment, but rather only to specify $O_3$ variability in the upper parts of the modeled atmosphere in the vicinity of the tropopause and lower stratosphere. We have further clarified this in the revised manuscript. Additionally, as shown in the comparisons of the results of the WRF simulations at 108, 36, and 12km resolutions at the locations of the ozonesonde sites examined in the study, the time-height variations in the PV fields are largely similar and so is the estimated altitude of the tropopause (i.e., altitude of 2PVU) and the eventual scaled $O_3$ levels above 100hPa at these locations. Consequently, we do not expect the inferences from the air mass characterization method to be adversely impacted by the choice of the 108km resolution. We of course do expect that the precise space and time variations in the transport characteristics of this simulated "stratospheric $O_3$" could potentially be different across these resolutions even though the identification of STT events is similar.

It is also important to note that the WRF simulations at all the 3 resolutions employed data assimilation wherein wind, temperature and water vapor fields were nudged to the 1 degree spatial and 6h temporal resolution NCEP/NCAR analysis fields. The nudging scheme has a strong influence on the simulated dynamics in the model's UTLS and is likely the primary reason why the estimated PV fields and simulated UTLS dynamics across the 3 resolutions are similar. It is conceivable that the use of finer resolution analysis fields in conjunction with a finer resolution model may result in larger differences across the model simulations, but such an investigation is clearly a much larger study in-itself.

Nowhere in the manuscript or our response do we convey that "*the amount of ozone transported down to the surface is resolution independent*". Comparisons of the WRF fields over the U.S. at the 3 resolutions however, indicate that for the current model configuration (physics options and data assimilation methodology), the $O_3$ fields in the UTLS resulting from the PV scaling parameterizations would be similar across the differing resolutions likely driven by the nudging to the 1-degree resolution analysis fields. One would expect that differences in 3D wind fields in the mid-troposphere to the surface and possible differences in the representations of clouds and boundary layer evolution would then result in differences in the amounts of the UTLS ozone that get transported to the surface. However, without conducting chemistry transport simulations at different resolutions across the hemispheric domain it is difficult to quantify what the likely impact on surface-level $O_3$ would be and whether the differences translate to appreciable model performance inferences which in turn would depend on the space and time averaging of the performance statistics measures considered. We do appreciate the reviewer's line of questioning but contend that the issue of determining optimal spatial resolution for representing STT processes and their eventual impact on modeled ground-level $O_3$ is not straightforward and that the choice is somewhat model application goal specific.

The determination of whether finer resolution systematically improves the model skill across space and time relative to available observations will also be influenced by the choice of model parameterizations employed for representing cloud and turbulent mixing as well as the quality and resolution of the analysis fields used in the WRF model data assimilation scheme. The systematic investigation of the effects of grid resolution, model parameterization, and quality and

resolution of the analysis and observation fields used in data assimilation on representation of STT processes is certainly a worthy but far more extensive research endeavor than the scope of the current study.

Finally, we would like to clarify that the intent of our study was not to advocate for the use of a 108km resolution and neither was the intent of the additional WRF simulations to suggest that representation of 3-D $O_3$ is resolution independent. Instead the latter were analyzed to simply ascertain whether our interpretation of STT events in context of the space and time scales examined were adversely impacted by the choice of the coarse resolution. Analysis of specific stratospheric intrusion events will likely benefit from finer resolution, improved model dynamics and assimilation of high space and time resolution measurements. We share the reviewer's view on the need for improved resolution (though we also stress both vertical and horizontal) and systematic investigation of its impact in improving modeled 3D $O_3$ distributions to aid practical assessments of the relative contributions of anthropogenic and natural sources to ground-level $O_3$.

We have carefully re-read the manuscript and have incorporated additional changes to help clarify the issues discussed above, and also correct instances with awkward phrasing and sentences. We trust that our response to the reviewer comments and the revisions incorporated help address the reviewer's concerns.

[revised manuscript text omitted]